# Characterizing the volatility and mixing state of ambient fine particles in summer and winter of urban Beijing

Lu Chen[1, 2], Fang Zhang[2*], Don Collins[3], Jingye Ren[1], Jieyao Liu[1], Sihui Jiang[1], Zhanqing Li[4]

[1]College of Global Change and Earth System Science, Beijing Normal University, Beijing 100875, China
5  [2]Environmental Science and Engineering Research Centre, School of Civil and Environmental Engineering, Harbin Institute of Technology (Shenzhen), 518055 Shenzhen, China
[3]Chemical and Environmental Engineering, University of California, Riverside, CA 92521, USA
[4]Earth System Science Interdisciplinary Center and Department of Atmospheric and Oceanic Science, University of Maryland, College Park, MD, USA

10  *Correspondence to*: F. Zhang (zhangfang2021@hit.edu.cn)

**Abstract.** Understanding the volatility of atmospheric aerosols is important for elucidating the formation of fine particles and to help determining their effect on environment and climate. In this study, the volatility of the fine particles (40, 80, 110, 150, 200 and 300 nm) is characterized by the size-dependent volatility shrink factor (VSF) for summer and winter in the urban area of Beijing using measurements of 15  a volatility tandem differential mobility analyzer (VTDMA). We show that there are two persistent aerosol volatility modes (one high-volatile and one less or non-volatile mode) present both in the summer and winter. On average, the particles are more volatile in the summer (with mean VSF of 0.3) than in the winter (with mean VSF of 0.6). Although the new particle formation (NPF) process requires low-volatile vapors to form molecular clusters and nuclei, the significant high-volatile mode around noontime on NPF 20  days indicates partitioning of volatile substances into the growing particles during summer. We further retrieve the mixing state of the ambient fine particles from the size-resolved VSF and find that the non-black carbon (BC) particles that formed from nucleation processes accounted for 52–69 % of the total number concentration in the summer. On the other hand, particles containing a refractory core that is thought to be BC-containing particles dominate and contribute 67–77 % toward the total number 25  concentration in the winter. The diurnal cycles of the retrieved aerosol mixing state for the summer further supports the conclusion that nucleation process is the main contributors to non-BC particles. In addition, the extent of aging of BC particles was characterized as the ratio of the BC diameter before and after heating at 300 ºC ($D_p/D_c$), showing that the average ratio of ~2.2 in the winter is higher than the average

of ~1.5 in the summer, which indicates that BC aging may be less efficient in summertime. This would result in differences in light absorption enhancement between the cold and warm seasons.

## 1 Introduction

The volatility of atmospheric aerosols affects their effect on climate, visibility and human health (Dzubay et al., 1982; Pöschl, 2005; Baklanov et al., 2016) by modulating mass concentrations and size distributions of aerosol particles via gas-particle partitioning. Aerosol measurement could be largely biased under different temperatures because of volatility (Meyer et al., 2000; Grieshop et al., 2006; Chen et al., 2010). In addition, volatility influences the partitioning of aerosols in gas and particle phases, and thus affects dry and wet deposition rates (Bidleman, 1988), chemical reaction mechanisms, and atmospheric lifetime (Huffman et al., 2009; Glasius and Goldstein, 2016). Therefore, it is important to study the volatility of aerosols in different regions and environments, including in polluted urban areas.

Laboratory and field measurements have shown that aerosol volatility is correlated with chemical composition of the particles, which is impacted by emission sources and atmospheric processes (Wehner et al., 2004; Yeung et al., 2014). Therefore, the volatility of fine particles varies greatly with time and location. A common measure of volatility is the shrink factor (VSF), which is defined as the ratio of the particle diameter after and before being heated and is measured by a volatility tandem differential mobility analyzer (VTDMA). The VSF can range from as low as 0 (completely volatile compounds) to 1 (indicating completely non-volatile substances, e.g. black carbon), reflecting heterogeneity in particle composition in diverse environments (Wang et al., 2017; Chen et al., 2020). In addition, the dependence of particle volatility on particle size is complex. For example, Wang et al. (2017) found that ambient aerosol volatility typically decreases as particle size increases in urban Beijing, whereas Levy et al. (2014) showed the opposite dependence on size near the California-Mexico border. Numerous studies have linked aerosol volatility to the presence and abundance of refractory carbonaceous compounds by inferring aerosol mixing state and the degree of aging from measured volatility (Wehner et al., 2009; Cheung et al., 2016; Zhang et al., 2016; Zhang et al., 2017; Chen et al., 2020). Mixing state has been found to vary significantly between clean and heavily polluted days (Wehner et al., 2009), with a corresponding decrease in the fraction of externally mixed black carbon (BC) particles from 37 % during

clean to 18 % during heavily polluted periods. Cai et al. (2017) showed that nearly all particles volatilized at about 300 ℃ in Okinawa, while 15–21 % did not in Pearl River Delta. Saha et al. (2018) found the non-volatile fraction in roadside aerosols was mostly externally mixed. The results from these studies show that aerosol mixing state and degree of aging may differ greatly under diverse ambient conditions.

Considering current uncertainties in assessing the radiative forcing of BC particles, which are largely due to uncertainties in the model treatment of BC mixing state, emissions, and removal processes (Cappa et al., 2012; Nordmann et al., 2014), an improved understanding of aerosol volatility and mixing state is hence needed.

Most previous studies in north China have focused on aerosol chemistry, sources, and transport
(Wang et al., 2010; Gao et al., 2011; Sun et al., 2016b), but few have linked the volatility and mixing state of the aerosol to its sources, formation and growth. In north China, severe haze usually occurs in the winter season, with extremely high $PM_{2.5}$, while in the summer it occurs much less frequently and with much lower $PM_{2.5}$, with the contrast resulting from influences of multiple factors such as regional and local emissions, particle formation, meteorology, and photochemistry. A comprehensive study on
investigating the aerosols volatility and mixing state in cold and warm seasons may help to elucidate the sources and chemical composition of aerosol particles.

In this study, the VTDMA system, which has been extensively employed in field observations (Sakurai et al., 2003; Wehner et al., 2009; Cheung et al., 2016; Wang et al., 2017), was used to measure size-resolved volatility at different heating temperatures during wintertime and summertime in Beijing.
At the maximum employed heating temperature of 300 ℃, volatile components tend to evaporate while leaving refractory materials such as BC and other non-volatile components (Cheng et al., 2009). Therefore, the VTDMA is also used to determine the mixing state of refractory carbonaceous particles (Wehner et al., 2009; Zhang et al., 2016; Chen et al., 2020). Here, we used the size-dependent VSF as a parameter to characterize the volatility behaviour of fine particles in cold and warm seasons of urban Beijing. In
addition, the mixing state of ambient fine particles in the summer, which is retrieved from the size-resolved VSF, is also compared to that in the winter. By contrasting the volatility and mixing state in the two seasons, this study is with aim of linking the aerosol particles volatile properties and mixing state to

its atmospheric chemical and physical processes under different ambient conditions in polluted urban areas.

## 2 Materials and Methods

### 2.1 Sampling site

We conducted two field campaigns from 28 January to 22 February 2019 and 15 July to 7 August 2019 at a site (Nanjiao; 39.81°N, 116.48°E) located southeast of urban Beijing. The sampling site is a basic meteorological observation station of the China Meteorological Administration (CMA), which hosts the most comprehensive meteorological observation instruments of all the CMA sites (Fig. 1). The main instruments used in this study were all placed there, which were set up and deployed in a container for measuring the physicochemical properties of aerosol particles simultaneously. The Fifth-ring beltway is nearby the sampling site with no major industrial pollution sources nearby.

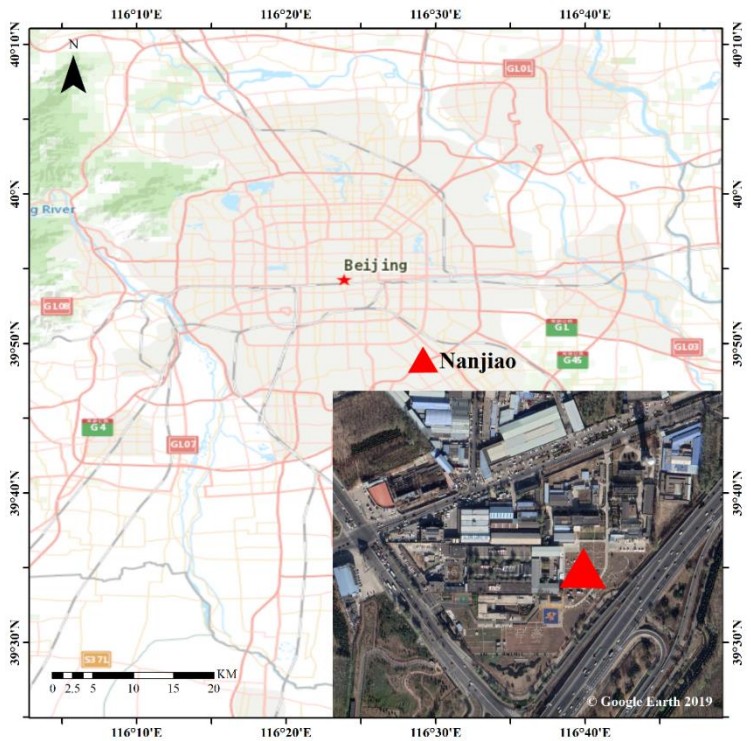

**Figure 1.** The map of the area surrounding the sampling site (Nanjiao) and true colour image of the observation station. This map was made by ArcGIS (http://www.arcgis.com/index.html#). The true colour image of the observation station was from © Google Earth 2019.

## 2.2 Instruments and measurements

The volatility and mixing state of fine particles were measured with a VTDMA (Fig. S1). Wang et al. (2017) provided a brief description of the custom-made VTDMA system. Here, we give more details. The instrument mainly consists of the following seven parts: neutralizer, the first differential mobility analyzer ($DMA_1$), temperature control module, the second DMA ($DMA_2$), water-based condensation particle counter (WCPC, TSI model 3787), auxiliary components (electric steering valve, vacuum pump, proportional valve, etc.), and software for control and data acquisition. During the measurement, ambient aerosols were first sampled by a $PM_{2.5}$ inlet and subsequently passed through a Nafion dryer that reduced the sample flow relative humidity (RH) to below 30 %. The dried aerosols were then directed through a neutralizer to neutralize the charge carried by the particles and entered the $DMA_1$ to produce quasi-monodisperse aerosols by setting the voltage. The dry diameters ($D_p$) selected in this study were 40, 80, 110, 150, 200 and 300 nm respectively. The selected quasi-monodisperse particles went either to the WCPC to obtain particle counts or through the heating tube for volatility measurements, sequentially at 80, 150, 200 and 300 ℃. Here, we focus on the data derived at 300 ℃. After heating, the sample flow entered the $DMA_2$ to scan the multi-dispersed particles, and finally entered the WCPC to get the particle size distributions after heating, thereby obtaining the volatility shrink factor measured distribution function (VSF-MDF). Through inversion, the volatility shrink factor probability distribution function (VSF-PDF) was further obtained. The VSF-PDF was retrieved based on the $TDMA_{inv}$ algorithm developed by Gysel et al. (2009). The scans in which the temperature between the two DMAs was not increased were used to define the width of the transfer function. The residence time in the heated region was 2.4 s (Cheung et al., 2016). Compared with that of 0.3 to 1.5 s for other VTDMA systems (e.g., Brooks et al., 2002; Philippin et al., 2004; Villani et al., 2007; Jiang et al., 2018), the residence time in this VTDMA is sufficient for the volatile materials to be effectively vaporized. The relative humidity was verified periodically with ammonium sulfate during the measurement period. The result of the verification is shown in Fig. S2. It shows that the deliquescence RH of ammonium sulfate measured by H/V-TDMA is ~78 %, which is consistent with the results reported by Badger et al (2006) and Tan et al (2013), indicating the RH measurement of this system is accurate. In addition to the volatility system, some other auxiliary instruments were used for simultaneous observation, including an aethalometer (AE33, Magee

Scientific) for measuring the mass concentration of BC, and a scanning mobility particle sizer (SMPS) for measuring the particle number size distribution (PNSD) of aerosols. Before the field measurement, all instruments used were calibrated to ensure the data obtained during the study period were accurate and reliable. Detailed calibration process of these auxiliary instruments can be found in Wu et al (2020). The meteorological variables from the meteorological observation station were also used, including ambient temperature (T), RH, wind direction (WD), and wind speed (WS).

**2.3 Method for retrieval of the mixing state of BC**

In this study, we use the abbreviations Ex-BC, In-BC, and Non-BC to denote externally mixed, internally mixed, and non-BC-containing particles, respectively. The number fraction of the completely volatile particles ($\Phi_{CV}$) was calculated as:

$$\Phi_{CV} = 1 - \frac{N_r}{N_{D_p} \cdot \eta_{D_p,T}} \tag{1}$$

Where $N_r$ is number concentrations of the residual particles after heating, $N_{D_p}$ is the number concentrations of DMA$_1$-selected particles before heating, and $\eta_{D_p,T}$ is the transportation efficiency of the sampled particles, which represents particle losses between DMA$_1$ and DMA$_2$ due to diffusion and thermophoretic forces (Philippin et al., 2004), and always determined at each particle diameter and heating temperature with sodium chloride (NaCl) particles in laboratory calibrations for that they do not evaporate even at high temperatures (Philippin et al., 2004; Cheung et al., 2016). In this study, $\eta_{D_p,T}$ at each particle size is determined from the number concentration of NaCl particles before and after heating at 300 °C (i.e. $\eta_{D_p,T} = N_r(NaCl)/N_{D_p}(NaCl)$). Due to the cubic shape of NaCl particles, a shape factor of 1.08 was used in the calibration process (Park et al., 2009; Hakala et al., 2017). The quantified number fraction of completely vaporized particles is shown in Fig. S3 and Fig. S4. We then classified the particles into three categories according to the measured VSF values (Wehner et al., 2009; Cheng et al., 2012):

- less-volatile (LV): VSF ≥ 0.82, considered to be Ex-BC;
- medium-volatile (MV): 0.45 ≤ VSF < 0.82, considered to be In-BC;
- high-volatile (HV): VSF < 0.45, considered to be Non-BC.

The VSF-PDF ($c(VSF, D_p)$) was normalized as $\int c(VSF, D_p)dVSF = 1$. Then, the number fraction ($\Phi_i$) for each volatile group with the boundary of $[VSF_{start}, VSF_{end}]$ is defined as:

$$\Phi_i = \left[\int_{VSF_{start}}^{VSF_{end}} c(VSF, D_p)dVSF\right] \cdot (1 - \Phi_{CV}) \tag{2}$$

Where $i$ = Ex-BC, In-BC or Non-BC. It is worth noted that, when $i$ = Non-BC, those particles that completely evaporate were assumed to be included in the HV mode (considered as Non-BC), so $\Phi_{Non-BC}$ is calculated as:

$$\Phi_{Non-BC} = \left[\int_{VSF_{start}}^{VSF_{end}} c(VSF, D_p)dVSF\right] \cdot (1 - \Phi_{CV}) + \Phi_{CV} \tag{3}$$

The number concentrations ($N_i$) of In-BC, Ex-BC, and Non-BC from the VSF distributions combined with the total PNSD simultaneously measured by the VTDMA, are calculated as follows:

$$N_i = \Phi_i \cdot N_{total} \tag{4}$$

Here, $N_{total}$ is the number concentrations of ambient fine aerosol particles. Therefore, the actual number fractions of Ex-BC, In-BC and Non-BC particles before heating could be obtained.

The ratio of the BC diameter before and after heating at 300 ℃ ($D_p/D_c$) was used as a quantitative index to characterize the coating thickness (degree of aging) of BC-containing particles. Here, particles with diameters of 150 nm are presented as examples (Fig. 2), $D_p$ refers to the peak value of the DMA$_1$ selected size distribution, and $D_c$ refers to the peak diameter of residual particles after heating at 300 °C.

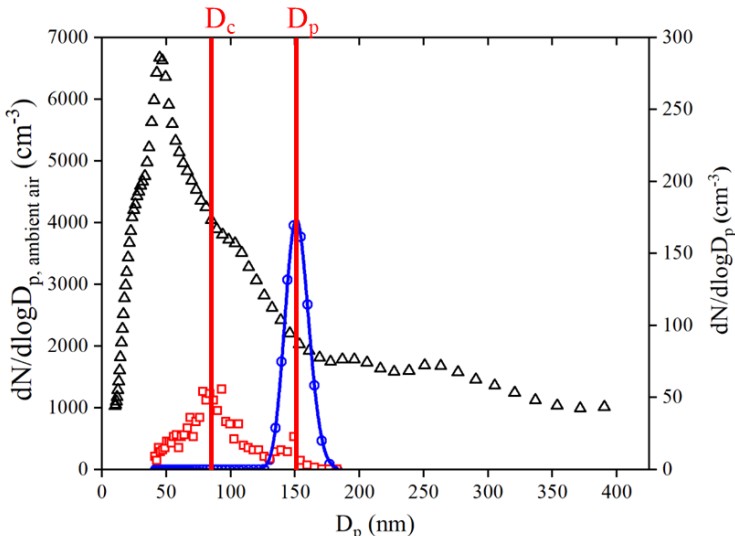

**Figure 2.** Particle number size distributions (PNSD) of ambient aerosols (in black), DMA$_1$-selected particles with $D_p$ equal to 150 nm (in blue), residual particles after heating at 300 °C (in red), and the fitting curves.

## 2.4 Uncertainty analysis of the retrieval method

Because this study investigates only the fine mode particles below 300 nm, refractory components that are present mostly in coarse mode particles, e.g. dust and sea salt, are expected to be negligible. At around 300–350 °C, the refractory component in sub micrometre aerosols in continental and urban areas has been considered to be mainly BC and a small contribution by charred organic material, which is often negligible (Rose et al., 2006; Frey et al., 2008; Wehner et al., 2009). Therefore, in this study, the retrieval of the mixing state of BC is based on the assumption that the refractory component in sub micrometre aerosols consists mainly or solely of BC. This assumption might not be true that there are some other important non-volatile aerosol compounds in submicron aerosols, for instance, some extremely low-volatility organic material that does not evaporate even at 300 °C (Cappa & Jimenez, 2010; Häkkinen et al., 2012; Poulain et al.,2014; Zhang et al., 2016; Wang et al., 2017).

To investigate the composition of the refractory component and verify whether they consist mainly of BC, we first quantify the bulk mass concentration of these non-volatile material. For the calculation, the number concentrations of the residual non-volatile particles at each size (40, 80, 110, 150, 200 and 300 nm) is calculated by integrating the residual PNSD of each selected particle size that directly

measured by VTDMA at the temperature of 300 °C. Then, the size resolved mass concentration of the residual non-volatile particles was calculated by assuming the particles are spherical and with a density of 1.6 g cm$^{-3}$ (Häkkinen et al., 2012; Poulain et al., 2014). Finally, by fitting the size-resolved mass concentration and integrating the fitted curves, the bulk mass concentration of non-volatile particles was retrieved. The retrieved bulk mass concentration of non-volatile particles was correlated with the BC mass

concentrations measured by AE33, which are shown in Fig. 3. The calculated non-volatile particle mass concentration and the measured BC concentration correlated well, with slope of 1.02. The mass fraction of non-volatile compounds except BC was further evaluated, which accounted for ~1.8 %. Consequently, at 300 °C, contribution of non-volatile OA factors is expected to be quite negligible (< 5 %). This result suggests that BC can explain almost mass fraction of the non-volatile material in this study. In addition,

we also compared the mean VSF measured at 200 and 300 °C, results show that the VSF values varied greatly under different heating temperatures especially for large particles (Fig. S5), hence some studies (Xu et al., 2016; Cappa & Jimenez, 2010) obtained considerable non-volatile OA at 250 °C may differ from the fractions at 300 °C.

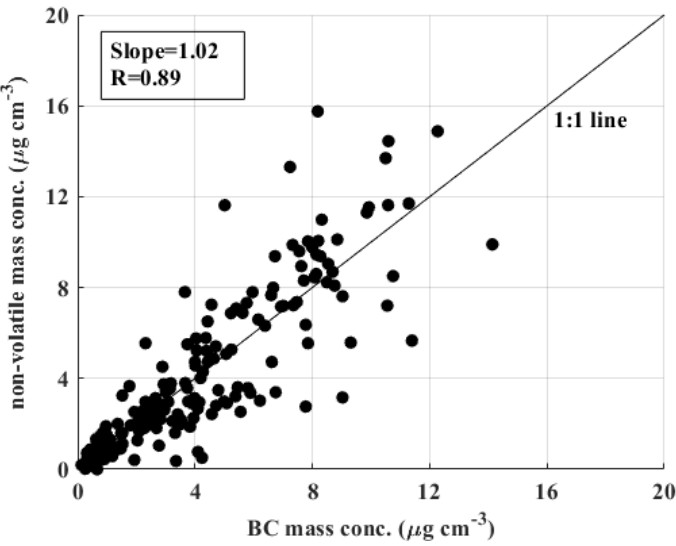

**Figure 3.** Black carbon (BC) mass concentration measured by AE-33 vs. the non-volatile mass concentration estimated from the VTDMA for winter 2019 periods. Estimation of the non-volatile mass concentration was made assuming a density of 1.6 g cm$^{-3}$.

To further verify the reliability of the retrieved results, the number fraction of Ex-BC and In-BC for 200 nm particles calculated from the VTDMA is compared with the measurements by single particle aerosol mass spectrometer (SPAMS), as shown in the Fig. 4. The comparison can only be confined to the size of 200 nm because which is the lower limit of the measured size for SPAMS (Bi et al., 2015). It exhibits that the variations of number fractions of both the Ex-BC and In-BC particles retrieved from VTDMA are well consistent with that measured by SPAMS, confirming that the method is reliable for deriving the mixing state of BC during the campaign in urban Beijing.

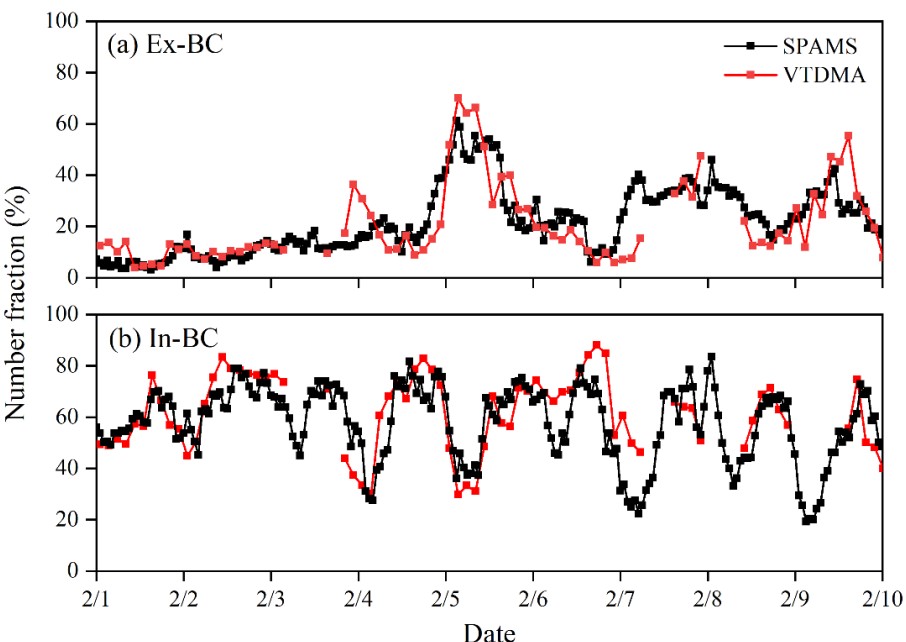

**Figure 4.** Time series of number fraction of **(a)** Ex-BC particles and **(b)** In-BC particles measured by SPAMS (in black) and calculated from VTDMA (in red), the 200 nm particles from VTDMA and SPAMS are chosen for comparison (1 February–10 February 2019).

## 3 Results and discussion

### 3.1 Time series of the VSF-PDF

Figure 5 shows the time series of the meteorological parameters, PM$_{2.5}$ concentration, and the VSF-PDF during the winter and summer campaign periods. The T and RH display diurnal cycles. The average T and RH are 0.3 °C and 35 % during the winter and 28.3 °C and 69 % during the summer. The dominant

winds at the site are from the north in the winter and from the south in the summer. The wind speed during
the two field campaigns ranged from 2 to about 6 m s$^{-1}$. More polluted episodes with PM$_{2.5}$ concentration
of $> 200$ μg m$^{-3}$ were more frequent in the winter than in the summer, when the PM$_{2.5}$ concentration was
$< 100$ μg m$^{-3}$ on most of the observed days. Figures 5d and 5e display time series of measured VSF-PDF
for DMA$_1$-selected particle sizes of 40 and 150 nm during the two periods. In the summer, the VSF
distributions of 40-nm particles were almost always bimodal, with a non-volatile mode (the VSF was
approximately equal to 1) and a high volatile mode (with VSF of about 0.2-0.5). As stated above, we
attribute the non-volatile group to refractory BC particles. In the winter, the VSF-PDF was bimodal only
occasionally, and mostly on polluted days, which could be caused by changes in meteorology and
enhanced primary BC emissions during polluted days in the winter. For the 150-nm particles, the
distributions were generally unimodal, with VSF of about 0.3-0.6 in the winter, but were almost bimodal
with a non-volatile mode and a high-volatile mode in the summer, indicating the mixing and aging of the
primary particles during growth to larger sizes during the winter sampling periods. The VSF in the
summer fluctuated a bit less than that observed in the winter, which will be further discussed in the
following section.

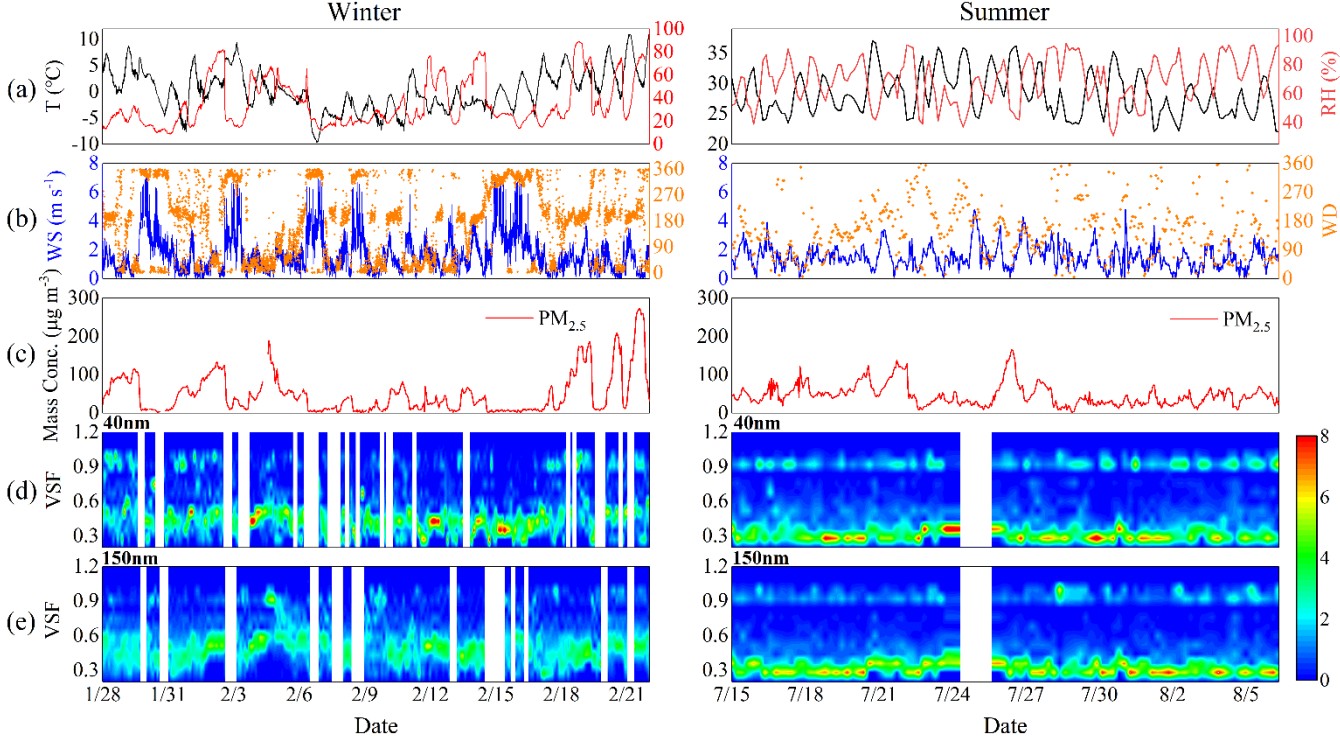

**Figure 5.** Time series of **(a)** ambient temperature (T) and relative humidity (RH); **(b)** wind direction and wind speed; **(c)** mass concentration of PM$_{2.5}$; **(d-e)** volatility shrink factor distributions (VSF-PDF) for 40- and 150-nm particles at T = 300 °C during the winter (left) and summer (right) periods.

### 3.2 Comparison of the VSF-PDF between summer and winter and trajectory analysis

Comparison of the average VSF-PDF distribution for all measured dry particle sizes during the winter and summer period is illustrated in Fig. 6. The mean VSF-PDFs are bimodal, with one high-volatile and one less- or non-volatile mode both in the summer and winter. But the mean VSF-PDFs possess an HV mode in the summer, generally with peak values of ~0.2, while an MV mode is present across the size range in the winter (with peak values of 0.45-0.65), indicating higher volatility of the aerosol particles in the summer. In addition, the modes in the VSF-PDFs for the different dry sizes are much broader in the winter than in the summer, reflecting greater heterogeneity in chemical composition.

Our results from Beijing are consistent with those reported previously for other urban environments, with Kuhn et al. (2005), Cheung et al. (2016), Cai et al. (2017), and Jiang et al. (2018) reporting that the VSF-PDFs of fine particles normally has both LV and HV/MV modes. The LV mode consists of non-

volatile particles, like BC, the MV mode is comprised of a mixture of volatile (e.g., organics) and non-volatile matter, and the HV mode generally consists of volatile materials that tend to evaporate when heated. In this study, the prominent MV mode in the winter suggests the dominant pathway for fine particle growth is coagulation and condensation of more volatile species (organics or inorganic salts) on non-volatile primary particles (e.g. BC), while the prominent HV mode in the summer especially for large particle size could be due to condensation of semi-volatile materials on nucleated particles (Riipinen et al., 2012).

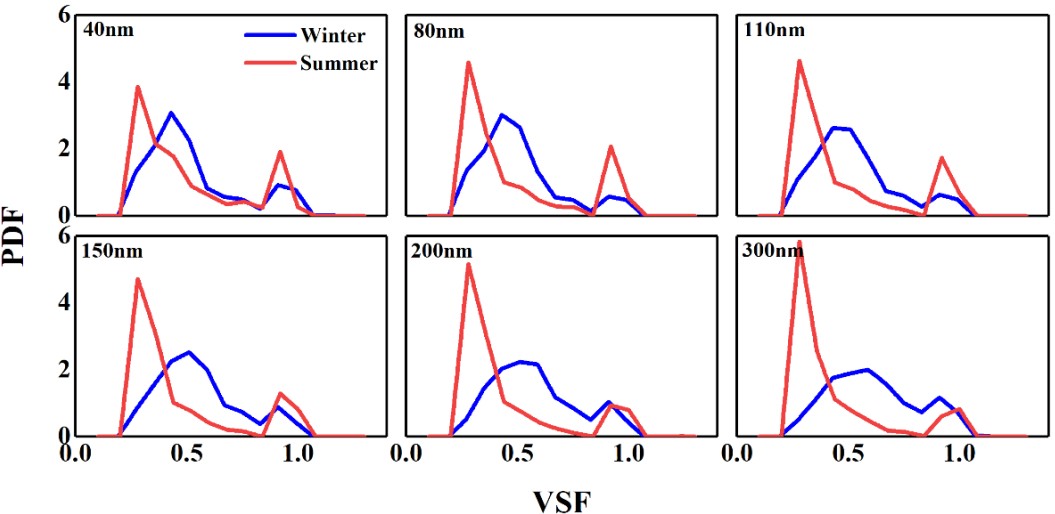

**Figure 6.** Average volatility shrink factor (VSF) distributions for different sizes (40-300 nm) during the winter (in blue) and summer (in red) observations.

Fig. 7 presents the 72-h back trajectories arriving at the sampling site during the two periods from 00:00 to 23:00 LT calculated applying the TrajStat software (Wang et al., 2009) (Fig. 7a), and the size-resolved mean VSF ($VSF_{mean}$) of the corresponding cluster during the winter and summer periods (Fig. 7b). In the winter, the air masses were categorized into five clusters and with prevailing northerly wind. The northwest clusters (C2 and C3) were predominant, which associated with the high $PM_{2.5}$ concentrations (Wang et al., 2015). The $VSF_{mean}$ of small particle size among different clusters shows greater variability than that of large size, implying more diverse sources of small particles. In the summer, the air masses were classified into six clusters, with prevailing southerly wind (C1 and C3). It shows that the $VSF_{mean}$ values in both the winter and summer were independent on the variations of trajectories, or

the impacts of regional transportation on volatility of the fine aerosol particles are complex. Obviously, the seasonal differences in VSF$_{mean}$ are more significant than that of among different clusters especially for larger size particles.

### 3.3 Comparison of diurnal variation of particles volatility between summer and winter

To obtain further insights into the effect of the formation and growth of particles on their volatility, we compare the diurnal variations of the observed mean VSF and VSF-PDF between the summer and winter (Fig. 8). Here, the VSF-PDFs of 40- and 150-nm particles are shown. Although the local primary emission sources (e.g. vehicles, cooking) can contribute to the pre-existing background small particles in urban environments, the fraction was probably lower during the new particle formation (NPF) events. It was reported that the ultrafine particles during clear days (e.g. NPF days) in urban Beijing were primarily from secondary formation (Wang et al., 2018), and the pre-existing particles are predominant in accumulated mode. During the summer, low VSF during the daytime (08:00–18:00 LT) and high VSF during the nighttime were observed for 40 nm and 150 nm particles (Fig. 8a). Accordingly, the VSF-PDF shows the HV mode dominated around noontime and early afternoon and the LV mode dominated during nighttime (Figs. 8c and 8e). The diurnal variation is more evident for small particles (e.g. 40 nm) than for larger particles (e.g. 150 nm). The diurnal variations illustrate that particles are more volatile during the daytime than at night, with VSF decreasing dramatically after ~10:00 LT. Fig. 8g displays the mean

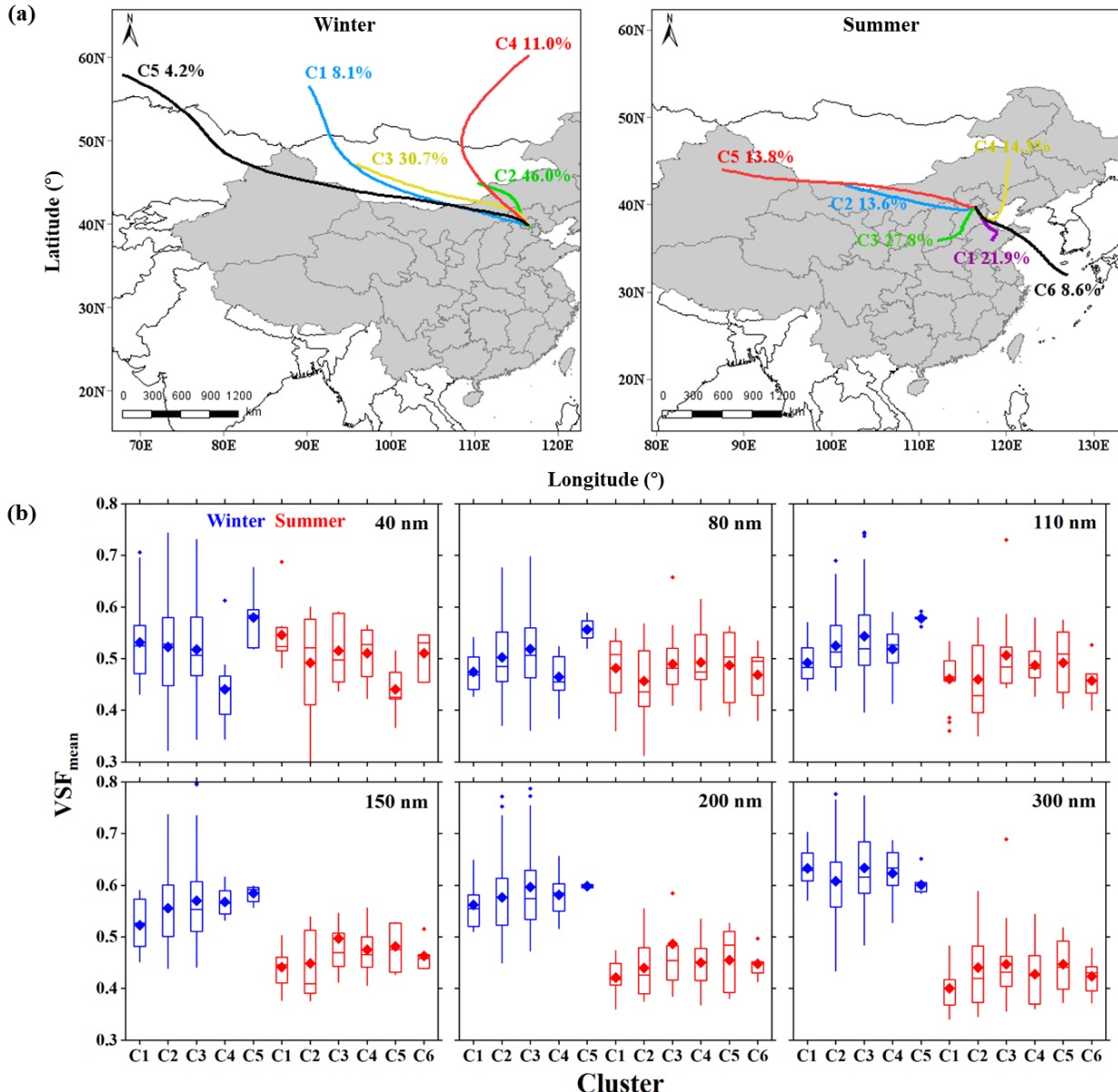

**Figure 7. (a)** The 72-h back trajectories arriving at the Nanjiao site during the winter (left) and summer (right) periods. C1-C6 represent Cluster 1-Cluster 6 respectively. The percentages present the relative occurrences. **(b)** Box diagram for the mean volatility shrink factor (VSF$_{mean}$) of all selected diameter particles (40-300 nm) from different clusters during the winter (blue) and summer (red) periods. The horizontal line in the block diagram

represents the median, the diamond represents the mean, the upper and lower borders represent the 25th and 75th percentiles, and the upper and lower borders of the dotted vertical line represent the 10th and 90th percentiles.

diurnal variations of PNSDs in the summer. During the summer sampling periods, NPF events took place frequently, with 10 NPF events occurred (Fig. S6). The NPF events almost all started at around 10:00 LT. After the starting of NPF, the volatile mode in VSF-PDF was obviously enhanced, corresponding to significant decreases of the mean VSF values. Although the NPF process requires low-volatile vapors to form molecular clusters and nuclei (Ehn et al., 2007), the significant high-volatile mode around noontime on NPF days indicates partitioning of volatile substances into the growing particles in the summer. Previous study also showed that the growth process of the nucleated particles primarily formed non-refractory sulfate and organics in Beijing (Wehner et al., 2009). Wu et al (2017) also observed that a clear decrease in VSFs for 30- and 50-nm particles in rural area of north China during the NPF events, indicating that volatile compounds could be produced during the growth process of newly formed particles. However, some earlier campaign measurements that were conducted in various atmospheric environments, such as urban (Sakurai et al., 2005), and forest (Ehn et al., 2007) showed that the volatility of newly formed particles varied with the atmospheric environments, indicating distinct particle growth mechanisms. In addition, during daytime atmospheric aging processes facilitated the mixing of primary particles (e.g. BC) with secondary species, leading to the transformation of externally mixed particles to internally mixed particles. In the evening and the early morning, the number fraction of LV-mode particles increased because of increased emissions of refractory particles (like BC) from traffic and other primary sources, coupled with slower particle aging and weaker vertical mixing that concentrates the externally mixed BC close to the surface (Zhang et al., 2016).

Compared with that in the summer, there was little diurnal variation in VSF during the winter period (Fig. 7b), and an MV mode was present in the VSF-PDFs for both 40- and 150-nm particles during the daytime (Figs. 8d and 8f). This is likely because of weakened photochemistry during the daytime in the cold season, when fewer NPF events were observed (Fig. 8h). In addition, the number fraction of the LV mode for both 40- and 150-nm particles is much lower during the winter (Figs. 8c-f). This may reflect the fact that BC particles can be coated and aged quickly through heterogeneous reactions of VOCs and other precursor gases (like $SO_2$ and $NO_x$) (Zhang et al., 2020), which are usually more concentrated during polluted days in the winter (Sun et al., 2016a). However, the aging process is expected to be slowed in the summer when anthropogenic precursor concentrations are lower. Such an explanation is reasonable

and can be supported by the observed thicker coating layer in the winter, as characterized by $D_p/D_c$ (shown in Fig. 12, see Sect. 3.6).

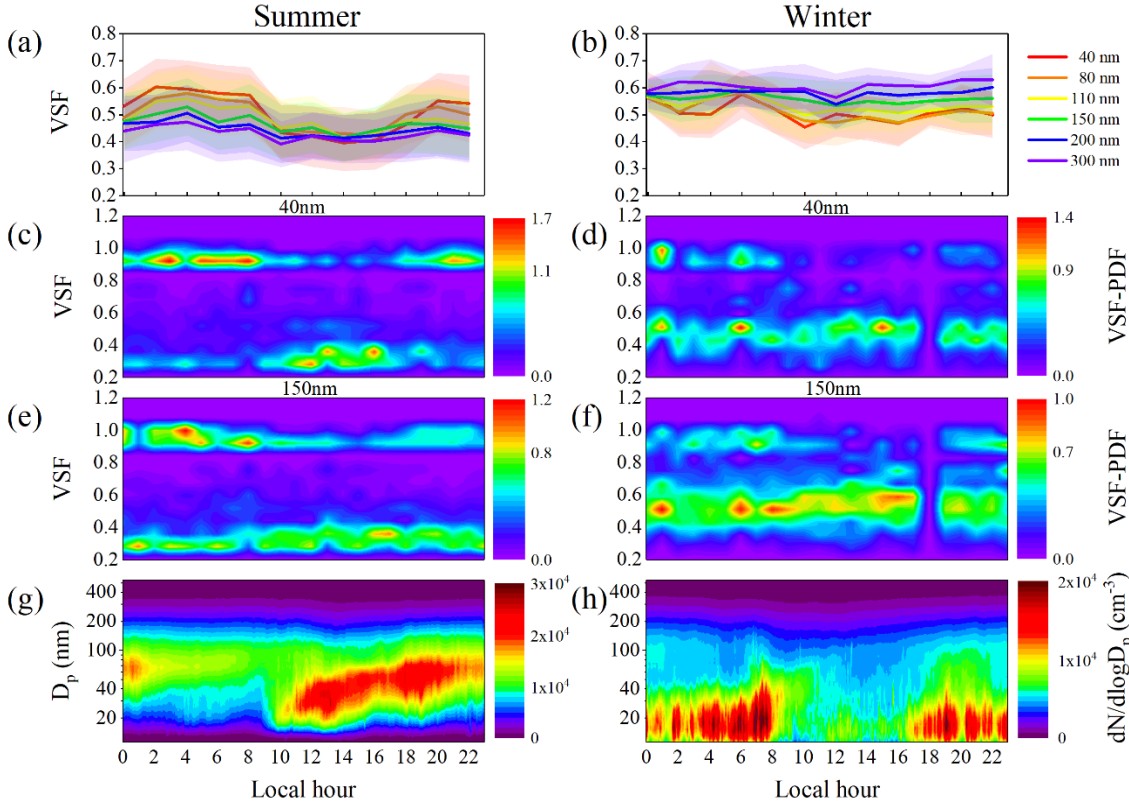

**Figure 8.** Diurnal variation of **(a-b)** mean VSF for all measured dry particle sizes, **(c-f)** mean VSF-PDF for 40- and 150-nm particles, and **(g-h)** mean particle number size distribution during the summer (left) and winter (right)
periods. The shade regions in (a-b) denote the standard deviations.

## 3.4 Number concentrations and fractions of Non-BC, In-BC, and Ex-BC

To study the aerosol mixing state, we retrieved the number concentrations of Non-BC, In-BC, and Ex-BC from the VSF data (Fig. 9). The time series of number concentrations and fractions of 150-nm particles are presented in Figs. 9a and 9b (see Figs. S7 and S8 for the time series of other sizes). Both in
the summer and winter cases, large temporal variations in the number concentrations and fractions of Non-BC, In-BC, and Ex-BC were observed, which reflects influences from both local primary urban emissions and addition of secondary species. In the summer, most particles were Non-BC (with mean

number fraction of ~67±9 %) because of large contributions from particles formed through condensation of semi-volatile materials, while In-BC accounted for a small proportion (~23 %). In contrast to the summer case, the majority of particles were In-BC in the winter, with a number fraction of ~55±11 %, reflecting efficient aging and coating of BC particles. Large Non-BC number concentrations and fractions were also observed by Zhang et al. (2017) in the summer at a site in Xianghe (a suburb site close to urban Beijing), where most ambient aerosol particles are Non-BC, with only 7–10 % of particles containing BC. The Ex-BC represents the smallest proportion both in the winter and summer, with number fractions of ~13±10 % and ~10±7 % respectively, suggesting rapid aging and mixing of freshly emitted BC with other species.

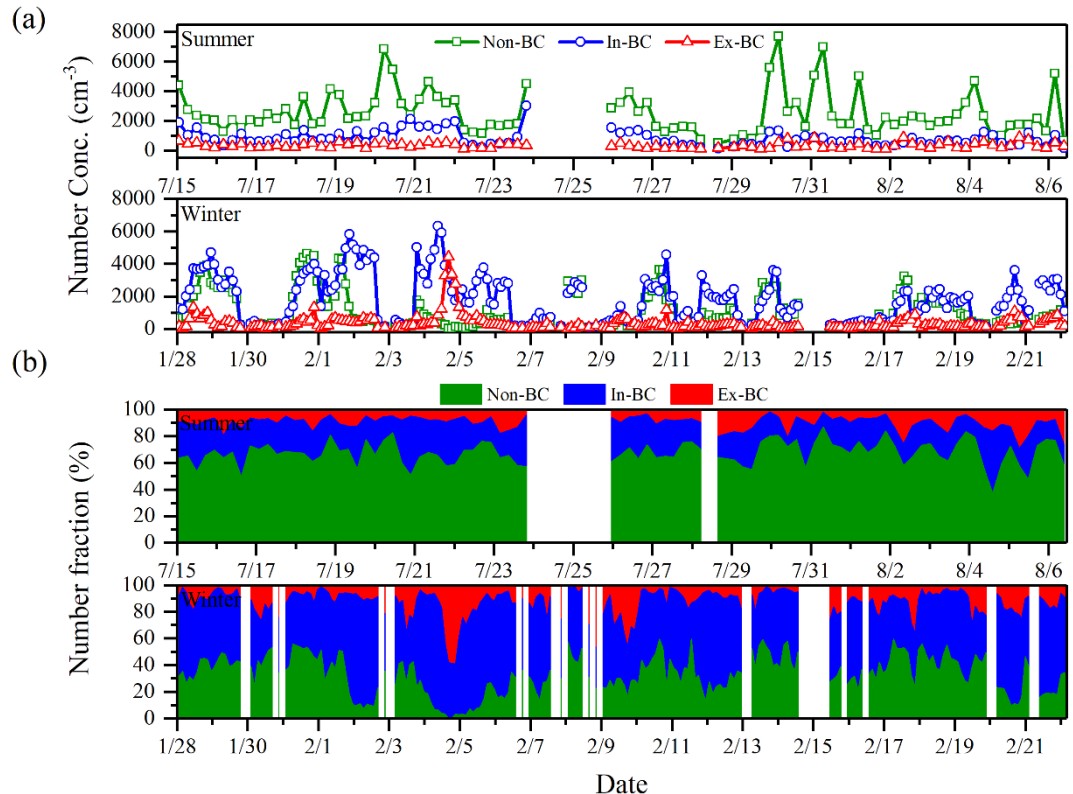

**Figure 9.** Summer (top) and winter (bottom) time series of **(a)** number concentrations and **(b)** number fractions of Non-BC (in green), In-BC (in blue), and Ex-BC (in red) of 150-nm particles.

Figure 10 illustrates the size dependence of the total number fractions of the Non-BC, In-BC, and Ex-BC particles during the summer and winter periods. The number fractions of Non-BC and In-BC are

highly size-dependent. In the summer, the number fraction of Non-BC increased with increasing particle size, accounted for 52–69 % of the total number concentration, further demonstrating the large contribution of condensation of semi-volatile materials to the total concentration of particles and the

absence of refractory materials (like BC) participating in their growth. While in the winter, the number fraction of In-BC increased and the number fraction of Non-BC decreased with increasing particle size, and BC-containing particles (including Ex-BC and In-BC) dominate and contribute 67–77 % toward the total number concentration, which again suggests that the particles in this size range result from growth of non-volatile primary aerosols (e.g. BC) through addition of more volatile components from coagulation

and condensation.

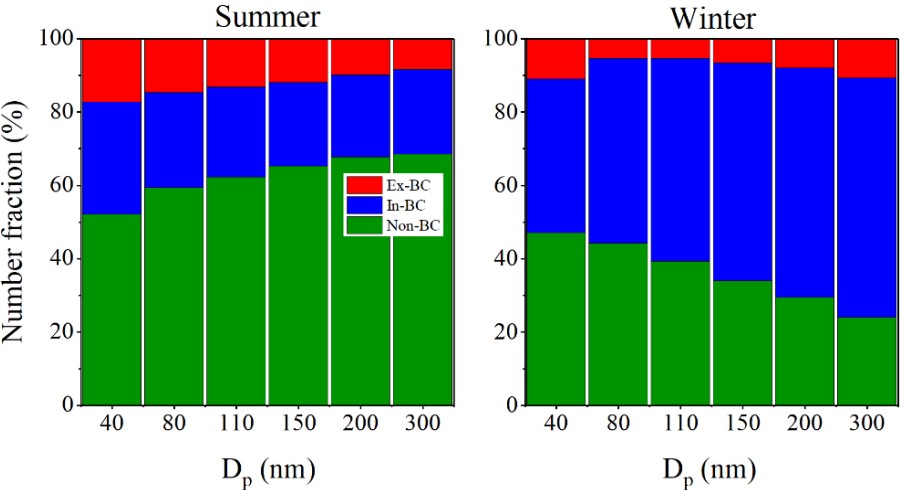

**Figure 10.** Size-resolved, campaign-averaged number fractions of Non-BC, In-BC, and Ex-BC during the winter (left) and summer (right) periods.

In summary, during the winter most ambient aerosol particles were BC-containing, suggesting that

BC particles are a dominant component in urban Beijing. In the summer, however, BC-containing particles contributed much less (only 31–48 %) toward the total number concentration in the measured size range, while Non-BC particles originating from condensation of semi-volatile materials are the dominant particle type.

### 3.5 Diurnal variations of retrieved number concentrations and fractions of Non-BC, In-BC, and Ex-BC

Figure 11 shows the diurnal variations of number concentrations and fractions of Non-BC, In-BC and Ex-BC of 40- and 150-nm particles for the summer and winter cases. Diurnal variations are evident for 40 nm and 150 nm Non-BC particles. In the summer, there is a continuous increase in the number concentration and fraction of Non-BC from 10:00 to 14:00 LT, reflecting the impact of new particle formation and growth. The increase concentration/fraction for 150-nm Non-BC particles extended into the late evening (23:00 LT) as particles that nucleated earlier in the day continued to grow into that size range. Compared to the summer, the Non-BC concentration/fraction in the winter was lower during daytime and higher during nighttime, with the average number concentration of ~1470 cm$^{-3}$ during nighttime (21:00-05:00 LT), corresponding to number fraction of ~40 %, but only ~942 cm$^{-3}$ (~30%) during daytime (06:00-20:00 LT). This illustrates that the diurnal changes in planetary boundary layer (PBL) dominates the diurnal patterns of Non-BC in cold season when the sources (e.g. nucleation) of Non-BC are insignificant.

The diurnal cycles for In-BC are insignificant both in the summer and winter. A slight increase in the number concentration of In-BC is observed during the daytime in the summer for 150-nm particles, likely resulting from rapid photochemical aging that converts the Ex-BC into In-BC particles. However, in the winter, photochemistry and the rate of aging of Ex-BC is reduced, and therefore little or no increase of In-BC around noontime is observed. Instead, its number concentration decreases slightly in the early afternoon (around 14:00 LT) because of the increase in PBL height, resulting in dilution of aged particles with the less polluted air from higher up (Rose et al., 2006). The diurnal variations of number concentrations and fractions of Ex-BC have minima during the daytime and maxima at night for both the summer and winter, consistent with diurnal variations of the PBL height and patterns in urban traffic, which emits fresh Ex-BC particles. These results are in general agreement with previous observations in similar urban environments such as those described by Cheung et al. (2016) and Cheng et al. (2012).

In summary, the diurnal variation of aerosol mixing state reflected competition among the processes of photochemical aging, nucleation, local primary emissions, and changes of PBL height. In the summer, photochemical production and growth of particles are the main contributors to Non-BC particles. The

rates of aging of BC particles in the urban atmosphere may be similar throughout the day as suggested by the insignificant diurnal variations of In-BC concentration and fraction. In addition to local vehicle emissions and incomplete combustion of fossil fuel, the diurnal variations of Ex-BC concentration are largely impacted by variations of the PBL.

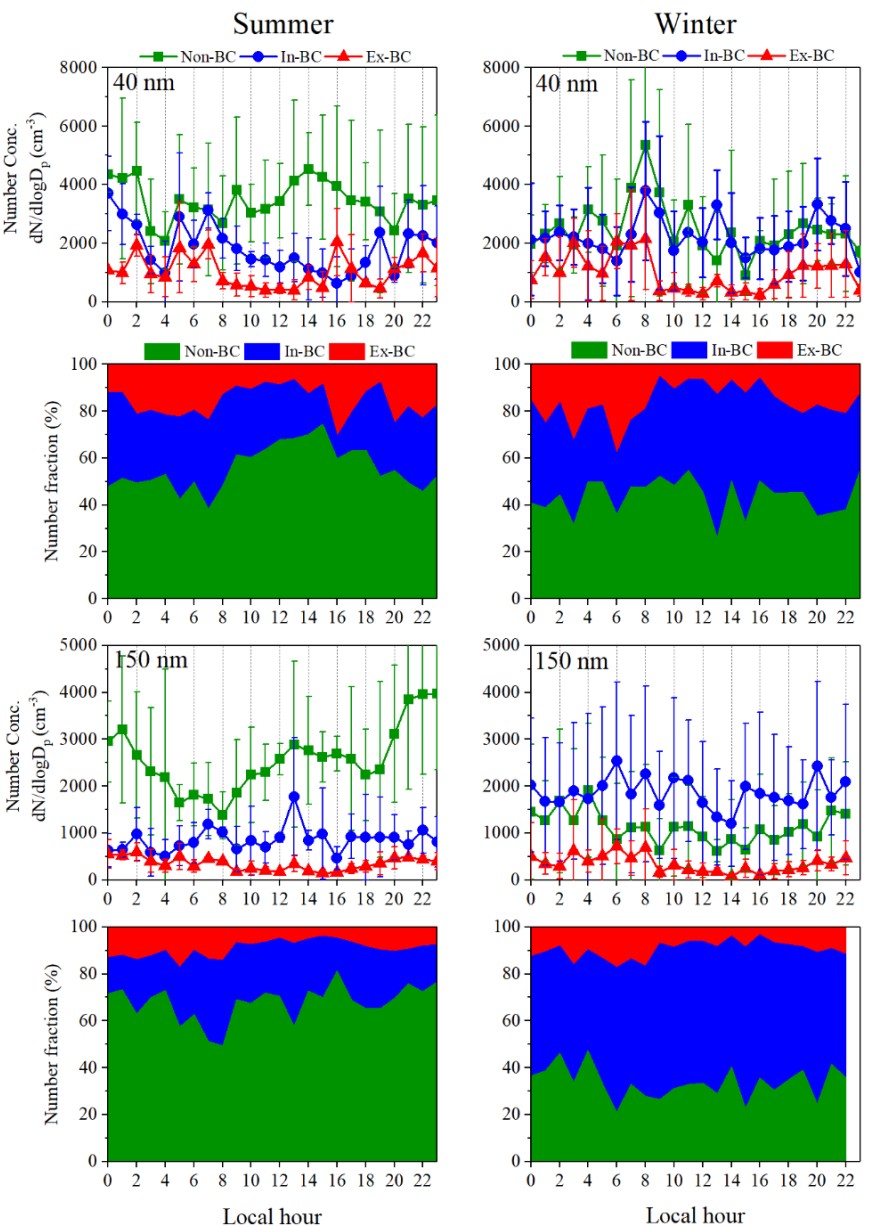

**Figure 11.** Average diurnal variations of retrieved number concentrations and fractions of Non-BC, In-BC, and Ex-BC of 40- and 150-nm particles in the summer (left) and winter (right) periods.

### 3.6 Coating thickness characterized by $D_p/D_c$ ratio

In the winter, most BC particles have $D_p/D_c$ ratios of 1.6-2.6 (Fig. S9). The large $D_p/D_c$ ratios suggest that the BC particles are thickly coated and likely have a compacted structure following atmospheric aging that results from additional emissions from the residential heating sector and favourable condensation because of the low temperature. While during the summer, the coatings on BC were thinner, with an average $D_p/D_c$ of 1.5 (Fig. 12). In addition, in the winter the $D_p/D_c$ ratio was highly dependent on particle size, with decreasing coating layer thickness with increasing particle size. In the summer, it was independent of particle size.

Our results are similar to those reported previously in similar urban environments (Wehner et al., 2004; Zhang et al., 2018; Liu et al., 2019). For example, Liu et al. (2019) found BC coating thickness was more variable in the winter than in the summer, and that the average coating thickness on BC particles was higher in the winter. Zhang et al. (2018) showed that the size-dependence of the $D_p/D_c$ ratio was associated with air pollution and indicated that the aging of smaller BC cores was more sensitive to air pollution levels.

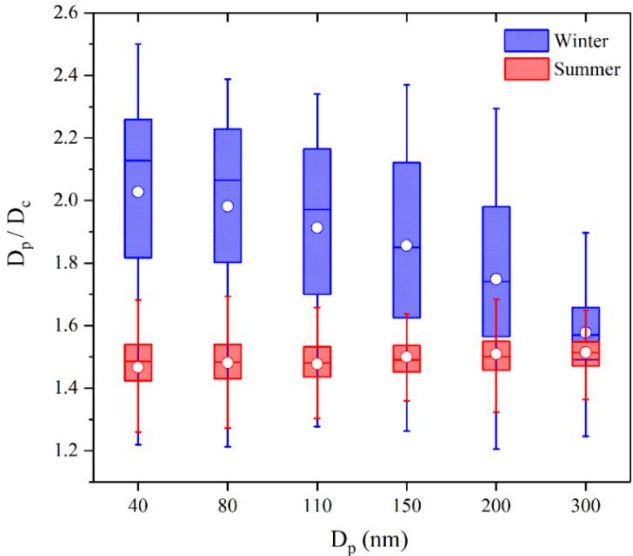

**Figure 12.** Size-dependent mean $D_p/D_c$ ratio (circular markers) of fine aerosol particles during the winter (in blue) and summer (in red) periods. The boxes show the 25th, 50th, and 75th percentiles. The extremities show the 5th and 95th percentiles.

Previous modelling studies have reported that coating materials on BC particles can significantly enhance the light absorption of BC via the lensing effect (Jacobson, 2001; Moffet and Prather, 2009; Lack and Cappa, 2010; Zhang et al., 2020). Thus, aging of BC-containing particles enhances their light absorption efficiency. However, how the aging and light absorption capability of BC particles will change under different ambient conditions remains unclear. The differences in coating thickness on BC particles we observed between summer and winter can help to parametrize the BC absorption enhancement in models.

## 4 Conclusion

In this study, the volatility of the fine particles is characterized as VSF and the results from wintertime and summertime are shown and compared. Results show that the measured VSF-PDF is almost always bimodal, with one high-volatile and one less- or non-volatile mode, both in the summer and winter. The mean VSF-PDF has a pronounced HV mode in the summer, generally with a VSF minimum at ~0.2, while in the winter a broad MV mode spans much of the measured VSF range (with VSF minimum at 0.45-0.65), reflecting lower average volatility in the winter. Diurnal variations in VSF of 40-nm particles are evident only in the summer, with a prominent HV mode present around noontime and early afternoon and a dominant LV mode during nighttime. No such feature is evident in the winter measurements. The mixing state of ambient fine particles was calculated from the size-resolved VSF and the results for summer and winter compared and contrasted. On average, nucleation and particle growth results in a dominant population of volatile, Non-BC, particles, which account for 52–69 % of the total concentration in the measured size range in the summer. However, black carbon (BC)-containing particles contributed 67–77 % toward the total number concentration in the winter, indicating that BC particles are a dominant component of the wintertime Beijing. The diurnal cycles of the retrieved particle mixing state in the summer further show that rapid photochemical processing and nucleation are the main contributors to Non-BC particles. By analyzing the ratio of the BC particle diameter before and after heating at 300 ℃, we show that the BC particles are more thickly coated in wintertime than in summertime. The observed results on aerosols volatility and mixing state in this study can help understanding the formation and growth of fine particles and determining their effect on environment and climate.

*Data availability.* All data needed to evaluate the conclusions in the paper are present in the paper and/or the Supplementary Materials. Also, all data used in the study are available from the corresponding author upon request (zhangfang2021@hit.edu.cn).

*Author contributions.* F. Z. and L. C. conceived the conceptual development of the manuscript. L. C. directed and performed of the experiments with J. L., S. J., J. R., and F. Z. and L. C. conducted the data analysis and wrote the draft of the manuscript, and all authors edited and commented on the various sections of the manuscript.

*Declaration of competing interest.* The authors declare no competing interests.

*Acknowledgments.* This work was funded by NSFC research projects (41975174, 41675141), and the National Basic Research Program of China (2017YFC1501702), and BNU Interdisciplinary Research Foundation for the First-Year Doctoral Candidates (Grant BNUXKJC2126). We thank all participants in the field campaigns for their tireless work and cooperation.

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
