# Peer review of "Characterizing the volatility and mixing state of ambient fine particles in summer and winter of urban Beijing"

_Atmospheric Chemistry and Physics, 2021_

## Referee Comment (RC1)

This contribution presents size-dependent volatility properties of urban aerosols in Beijing during summer and winter time using VTDMA. This work tries to calculate and compare the number concentration of In-BC, Ex-BC and Non-BC particles from the particle volatility distribution after heating up to 300 °C for aerosols under the studied two seasons. In addition, the extent of aging of BC particles was characterized based on current datasets. Although the comprehensive dataset presented in the manuscript is interesting and may carry certain values for the scientific community, the major conclusion separating In-BC, Ex-BC and Non-BC seems to be poorly supported by current analysis. The mixing state of BC particles derived from VTDMA data is highly uncertain or even not correct, and the uncertainties were not carefully analyzed in the manuscript. I therefore do not recommend publication in ACP.

Major comments:

The major results of current manuscript were based on the assumption that in urban areas refractory component in sub micrometer aerosols consists mainly or solely of BC. This assumption might not be true that there are some other important non-volatile aerosol compounds in submicron aerosols, for instance, some extremely low-volatility organic material that does not evaporate even at 300 °C . Nowadays, too many studies (Cappa & Jimenez, 2010; Häkkinen et al., 2012; Poulain et al.,2014; Wang et al., 2017; Zhang et al., 2016) confirmed that besides BC, low-volatility oxygenated organic aerosols also existed in the non-volatile fractions in particles. Specifically, Xu et al. (2016) measured the chemical composition of PM1 after heating at a rural site (Detling, Kent) and found at 250 °C, OA has the largest contribution (~ 40 %) to the residual mass. Hence, based on aerosol volatility properties, you may not be able to characterize the mixing state of BC particles.

Then, it comes with my second comment regarding the method this manuscript used to category In-BC, Ex-BC and Non-BC particles as given in line 146-149. Even though this method has been used in previous studies as in Wehner et al. (2009) and Cheng et al. (2012), it is actually out of date and not recommended to be employed to classify the mixing state of BC-particles. With VSF > 0.82, ambient particles could be composed of only BC, but also In-BC coating with non-volatile organics. Therefore, using current classification, significant uncertainties may be introduced into the estimation of the number fraction of different BC-containing particles. We now rely on some other techniques, for instance, SP2, VTDMA-SP2, SP-AMS to study the mixing state of BC-containing particles.

Moreover, in line 161-162, you concluded your results were reliable for deriving the mixing state of BC by comparing with SPAMS results. However, Fig.2 in your previous publication (Chen et al., 2020) only compared the total BC-containing particles; separation of Ex-BC particles from In-BC particles was not performed. I did not see any other evidence that could support your conclusion here. If you have, please specify.

Large uncertainties using Eq. 1 in your manuscript to calculate completely volatile particles may arise. Transportation efficiency as I understand was determined by volatility measurements of NaCl particles or other non-volatile particles at 300 °C in your study. However, ambient aerosols with different volatility could evaporate differently, resulting in different particle size after heating. Therefore, you have to at least consider the number size distribution of particles after heating into the determination of the transportation efficiency. In other words, number concentration of completely volatile particles should be dependent on the volatility of your ambient particles, thus your Eq.1 should be revised as a function of your VSF or your volatility distribution after heating.

In section 3.3, you studied the effect of the formation and growth of particles on their volatility. However, your discussion and analysis is not thorough. How frequent of NPF events occurred during your studied seasons stated in your manuscript? Did you select the data of the days of NPF events to drive your conclusion, otherwise, how could the influence of other atmospheric processes be neglected? What are the sources of 40 and 150 nm particles? Your 40 nm particles could be newly formed particles or a mixture of newly formed particles after growth with pre-exsiting particles. Your analysis should guide the reader to a clear process level, otherwise your conclusion will be quite difficult for us to sink into. Moreover, are the plots in Fig. 4a-h the average or median values of the whole campaign or of the NPF days? I think Fig. 4g or Fig. 4h in your manuscript present a certain NPF day, how could you compare general patterns with a case event? How could we reader withdraw general information or conclusions by case studies?

In section 3.6, you characterized coating thickness by $D_P/D_C$ ratio. However, I did not find any description or definition about Dp and Dc. You obtained a volatility distribution of particles after heating. How did you get Dc from this distribution? The determination of Dc was quite difficult, even with a SP2, which could accurately measure the mass of BC. Zhang et al. (2016) gave a thorough discussion of the uncertainties associated with determining Dc from SP2 measurements due to the morphology effect. Similar discussion of the uncertainties in determining Dc should be given in the manuscript.

References:

Cappa, C. D. and Jimenez, J. L.: Quantitative estimates of the volatility of ambient organic aerosol, Atmos. Chem. Phys., 10, 5409–5424, doi:10.5194/acp-10-5409-2010, 2010.

Chen, L., Zhang, F., Yan, P., Wang, X., Sun, L., Li, Y., Zhang, X., Sun, Y., and Li, Z.: The large proportion of black carbon (BC)-containing aerosols in the urban atmosphere, Environmental Pollution, 263, 114507, https://doi.org/10.1016/j.envpol.2020.114507, 2020.

Häkkinen, S. A. K., Äijälä, M., Lehtipalo, K., Junninen, H., Back man, J., Virkkula, A., Nieminen, T., Vestenius, M., Hakola, H., Ehn, M., Worsnop, D. R., Kulmala, M., Petäjä, T., and Riipinen, I.: Long-term volatility measurements of submicron atmospheric aerosol in Hyytiälä, Finland, Atmos. Chem. Phys., 12, 10771–10786, doi:10.5194/acp-12-10771-2012, 2012.

Poulain, L., Birmili, W., Canonaco, F., Crippa, M., Wu, Z. J., Nordmann, S., Spindler, G., Prévôt, A. S. H., Wiedensohler, A., and Herrmann, H.: Chemical mass balance of 300 ∘C non volatile particles at the

tropospheric research site Melpitz, Germany, Atmos. Chem. Phys., 14, 10145–10162, doi:10.5194/acp-14-10145-2014, 2014.

Wang, Z., W. Birmili, A. Hamed, B. Wehner, G. Spindler, X. Pei, Z. Wu, Y. Cheng, H. Su, and A. Wiedensohler: Contributions of volatile and nonvolatile compounds (at 300°C) to condensational growth of atmospheric nanoparticles: An assessment based on 8.5 years of observations at the Central Europe background site Melpitz, J. Geophys. Res. Atmos., 122, 485–497, doi:10.1002/2016JD025581, 2017.

Xu, L., Williams, L. R., Young, D. E., Allan, J. D., Coe, H., Massoli, P., Fortner, E., Chhabra, P., Herndon, S., Brooks, W. A., Jayne, J. T., Worsnop, D. R., Aiken, A. C., Liu, S., Gorkowski, K., Dubey, M. K., Fleming, Z. L., Visser, S., Prévôt, A. S. H., and Ng, N. L.: Wintertime aerosol chemical composition, volatility, and spatial variability in the greater London area, Atmos. Chem. Phys., 16, 1139–1160, https://doi.org/10.5194/acp-16-1139-2016, 2016.

Zhang, Y., Zhang, Q., Cheng, Y., Su, H., Kecorius, S., Wang, Z., Wu, Z., Hu, M., Zhu, T., Wiedensohler, A., and He, K.: Measuring the morphology and density of internally mixed black carbon with SP2 and VTDMA: new insight into the absorption enhancement of black carbon in the atmosphere, Atmos. Meas. Tech., 9, 1833–1843, https://doi.org/10.5194/amt-9-1833-2016, 2016.

---

## Referee Comment (RC2)

General Comments:

Chen et al. conducted ambient measurements in Beijing in summer and winter from which they calculated size-dependent volatility shrinkage factors (VSF) and mixing states of urban aerosols and compared the volatility properties in different seasons. The measurements were conducted with the use of a VTDMA of which size-selected aerosols ranging from 40-nm to 300-nm were heated up to 300$^{\circ}$C. The non-volatile particles that remained in the particle phase upon heating up to 300$^{\circ}$C were assumed to be black carbon (BC) in the analysis. Although volatility analysis of ambient aerosols has been intensively studied in general, this manuscript presents measurement results in different seasons in north China. The dataset presented in the manuscript is overall comprehensive but could be more thorough when interpreting the results.

Specific Comments:

1. I have similar concerns about the major assumption of attributing the non-volatile composition in urban aerosols to be BC in your analysis, as already detailed by another referee. Please consider providing more information, such as chemical composition data, or data from other instruments, if available, to support your assumption, which is critical to your analyses and discussions thereafter.

2. Since the manuscript aims to characterize the volatility properties of urban aerosols and tries to link the properties to the source, formation and growth, the authors may consider adding more materials to enrich the discussion, such as air masses origins information and their effect on the aerosols' volatility and mixing states.

3. It is frequently mentioned throughout Section 3 about the impact of new particle formation (NPF) and the growth on the volatility properties of aerosols. Please consider providing more details, such as the number of NPF events in summer and winter, respectively, to give a clearer picture and support to your analysis. Furthermore, although NPF events occurred less frequently in winter, did it have similar impact on aerosols' volatility as that in summer?

4. In line 178 – 181, the authors mention that the distributions of VSF for 150-nm particles were generally unimodal in both summer and winter. However, from Fig. 2(e), it seems that 150-nm particles were generally bi-modal with a non-volatile mode and a high-volatile mode in winter.

5. Section 3.3 compares the diurnal variation of particles volatility between summer and winter based on the mean VSF and VSF probability distribution function (VSF-PDF) as illustrated on Fig. 4, yet I was lost from line 229 to 236 when the number fraction of the low-volatile mode is discussed. Is the discussion still based on Fig. 4 or other figures in the manuscript?

6. Fig. 5(a) presents the time series of the number concentrations of Non-BC, In-BC and Ex-BC 150-nm particles in summer and winter. While this work shall be the same as that presented in their previous publication (Chen et al., 2020), the number concentration of 150-nm particles in this manuscript seems to be different from that on Fig. 5(a) in Chen et al. (2020). The scale also looks different from that of other sizes as shown in Fig. S4 in the Supplement.

7. In line 285 – 288, the authors state: "non-BC concentration / fraction in winter exhibits a daily minimum and nightly maximum". It is not clear to me whether this observation is supported by Fig. 7. For example, for 40-nm particles, there seems to be a morning peak at 08:00 LT for non-BC concentration in winter. For 150-nm particles, I am not sure whether there was any significant diurnal cycle for non-BC concentration / fraction. Please further elaborate to support your analysis.

8. The analysis method for the ratio of BC diameter discussed in Section 3.6 should be added in Section 2.

Reference:

Chen, L., Zhang, F., Yan, P., Wang, X., Sun, L., Li, Y., Zhang, X., Sun, Y., and Li, Z.: The large proportion of black carbon (BC)-containing aerosols in the urban atmosphere, Environmental Pollution, 263, 114507, https://doi.org/10.1016/j.envpol.2020.114507, 2020.

---

## Author Comment (AC2)

**A point-by-point response to the reviewer**

**General Comments:**

Chen et al. conducted ambient measurements in Beijing in summer and winter from which they calculated size-dependent volatility shrinkage factors (VSF) and mixing states of urban aerosols and compared the volatility properties in different seasons. The measurements were conducted with the use of a VTDMA of which size-selected aerosols ranging from 40-nm to 300-nm were heated up to 300°C. The non-volatile particles that remained in the particle phase upon heating up to 300°C were assumed to be black carbon (BC) in the analysis. Although volatility analysis of ambient aerosols has been intensively studied in general, this manuscript presents measurement results in different seasons in north China. The dataset presented in the manuscript is overall comprehensive but could be more thorough when interpreting the results.

**Re:** We are grateful to the reviewer for your insightful comments, all of which have been considered carefully during the revision (a point-by-point response to the reviewer as follows). The result has been carefully analyzed accordingly to the reviewer's suggestions and other comments have also been addressed.

**Specific Comments:**

1. I have similar concerns about the major assumption of attributing the non-volatile composition in urban aerosols to be BC in your analysis, as already detailed by another referee. Please consider providing more information, such as chemical composition data, or data from other instruments, if available, to support your assumption, which is critical to your analyses and discussions thereafter.

**Re:** It is possible that there may be some other non-volatile compounds in submicron aerosols will not evaporate even at 300 °C. To investigate the composition of the refractory component and verify whether they consist mainly of BC, we quantify the mass concentration of these non-volatile material and correlate it with the measured BC mass concentrations by AE33, which are shown in Fig. R1. The calculated non-volatile particle mass concentration and the measured BC concentration correlated well, with slope of 1.02. The mass fraction of non-volatile compounds except BC was further evaluated, which accounted for ~1.8 %. Consequently, at 300 °C, contribution of non-volatile materials except BC is expected to be quite negligible (< 5 %). This result suggests that BC can explain almost the non-volatile mass fraction in this study. In addition, we compared the mean VSF measured at 200 and 300 °C, results show that the VSF values varied greatly under different heating temperatures especially for large particles (Fig. R2), hence some studies (Xu et al., 2016; Cappa & Jimenez, 2010) obtained considerable non-volatile OA at 250 °C may differ from the fractions at 300 °C.

Actually, the composition of these non-volatile residuals may vary spatially and temporally (Poulain et al., 2014). For example, black carbon is considered a major nonvolatile component in sub-µm PM in many studies (e.g., Pöschl, 2005; Frey et al., 2008; Birmili et al., 2009; Birmili et al., 2010). Frey et al. (2008) have demonstrated a good agreement between the mass concentration of BC and the mass concentration of nonvolatile particles upon heating at 300 °C in the VTDMA; Birmili et al. (2009) found linear relationships between the BC mass concentrations and the non-volatile volume concentrations for five different atmospheric measurement sites; Birmili et al. (2010) found the non-volatile aerosol material had a clear correlation with BC aerosols in polluted areas. Generally, BC constitutes a major part of the non-volatile mass concentration.

The reliability of retrieved Ex-BC is also verified by comparison with SPAMS data, see Fig. R3, both the overall temporal variations and proportion for Ex-BC particles measured by SPAMS and calculated from VTDMA using the retrieved method are consistent, confirming our method are reliable for retrieving mixing state of BC in the study periods. Some discussions and evidences about this issue have been included in the revised paper, see Lines 122-132 and Lines 172-177, as follows:

"The detailed correlations between mass concentration of non-volatile particles estimated from VTDMA and BC measured by AE-33 in this study are shown in Fig. S2. The total mass concentration of non-volatile particles was determined from the measured particle number size distributions after heating by assuming particle density of 1.6 g cm-3 (Häkkinen et al., 2012; Poulain et al., 2014). The calculated non-volatile particle mass concentration distributed on both sides of the line 1:1. We further evaluated the mass fraction of non-volatile compounds except BC, which accounted for ~1.8 %. This result suggests that BC can explain almost of the non-volatile mass fraction in this study. In addition, we compared the mean VSF measured at 200 and 300 °C, results show that the VSF values varied greatly under different heating temperatures especially for large particles (Fig. S3), hence some studies (Xu et al., 2016; Cappa & Jimenez, 2010) obtained considerable non-volatile OA at 250 °C may differ from the fractions at 300 °C."

"The retrieval result, which has been compared with the measurements by single particle aerosol mass spectrometer (SPAMS) in the Fig. S5 and in the previous study (Chen et al., 2020), both the overall temporal variations and proportion for BC-containing and Ex-BC particles measured by SPAMS and calculated from VTDMA using the retrieved method are consistent, confirming this method is reliable for deriving the mixing state of BC."

**Figure R1.** Black carbon (BC) mass concentration measured by AE-33 vs. the non-volatile mass concentration estimated from the VTDMA for winter 2019 periods. Estimation of the non-volatile mass concentration was made assuming a density of 1.6 g cm-3.

---

## Author Response (AR1)

**A point-by-point response to reviewers**

Dear Editor,

We are very pleased to submit a revised manuscript entitled with "Characterizing the volatility and mixing state of ambient fine particles in summer and winter of urban Beijing" for possible publication in journal of ACP.

I'd like to thank you for your efforts and time on handling the paper. I also would like to thank the reviewers for their valuable comments and suggestions, all of which have been considered carefully during the revision (a point-by-point response to reviewers as follows). We believe all the comments from the reviewers have been addressed, and the paper have been greatly improved after the revision.

Yours sincerely,

Fang Zhang

On behalf of all authors

**Comments from the reviewers:**

**Reviewer #1**

This contribution presents size-dependent volatility properties of urban aerosols in Beijing during summer and winter time using VTDMA. This work tries to calculate and compare the number concentration of In-BC, Ex-BC and Non-BC particles from the particle volatility distribution after heating up to 300 °C for aerosols under the studied two seasons. In addition, the extent of aging of BC particles was characterized based on current datasets. Although the comprehensive dataset presented in the manuscript is interesting and may carry certain values for the scientific community, the major conclusion separating In-BC, Ex-BC and Non-BC seems to be poorly supported by current analysis. The mixing state of BC particles derived from VTDMA data is highly uncertain or even not correct, and the uncertainties were not carefully analyzed in the manuscript. I therefore do not recommend publication in ACP.

**Re:** We are grateful to the reviewer for your valuable comments and suggestions, all of which have been considered carefully during the revision (a point-by-point response to reviewer as follows). The reliability of the retrieved method has been carefully analyzed and other comments have also been addressed.

Major comments:

The major results of current manuscript were based on the assumption that in urban areas refractory component in sub micrometer aerosols consists mainly or solely of BC. This assumption might not be true that there are some other important non-volatile aerosol compounds in submicron aerosols, for instance, some extremely low-volatility organic material that does not evaporate even at 300 °C. Nowadays, too many studies (Cappa & Jimenez, 2010; Häkkinen et al., 2012; Poulain et al.,2014; Wang et al., 2017; Zhang et al., 2016) confirmed that besides BC, low-volatility oxygenated organic aerosols also existed in the non-volatile fractions in particles. Specifically, Xu et al. (2016) measured the chemical composition of PM1 after heating at a rural site (Detling, Kent) and found at 250 °C, OA has the largest contribution (~ 40 %) to the residual mass. Hence, based on aerosol volatility properties, you may not be able to characterize the mixing state of BC particles.

**Re:** The reviewer is right that there may be some other non-volatile compounds in submicron aerosols will not evaporate even at 300 °C, such as the extremely low-volatile organic material. To investigate the composition of the refractory component and verify whether they consist mainly of BC, we quantify the mass concentration of these non-volatile material and correlate it with the measured BC mass concentrations by AE33, which are shown in Fig. R1. The calculated non-volatile particle mass concentration and the measured BC concentration correlated well, with slope of 1.02. The mass fraction of non-volatile compounds except BC was further evaluated, which accounted for ~1.8 %. Consequently, at 300 °C, contribution of non-volatile OA factors is expected to be quite negligible (< 5 %). This result suggests that BC can explain

almost the non-volatile mass fraction in this study. In addition, we compared the mean VSF measured at 200 and 300 °C, results show that the VSF values varied greatly under different heating temperatures especially for large particles (Fig. R2), hence some studies (Xu et al., 2016; Cappa & Jimenez, 2010) obtained considerable non-volatile OA at 250 °C may differ from the fractions at 300 °C.

Actually, the composition of these non-volatile residuals may vary spatially and temporally. For example, black carbon is considered a major non-volatile component in sub-$\mu$m PM in many studies (e.g., Pöschl, 2005; Frey et al., 2008; Birmili et al., 2009; Birmili et al., 2010). Frey et al. (2008) have demonstrated a good agreement between the mass concentration of BC and the mass concentration of non-volatile particles upon heating at 300 °C in the VTDMA; Birmili et al. (2009) found linear relationships between the BC mass concentrations and the non-volatile volume concentrations for five different atmospheric measurement sites; Birmili et al. (2010) found the non-volatile aerosol material had a clear correlation with BC aerosols in polluted areas. Generally, BC constitutes a major part of the non-volatile mass concentration. Some discussions about this issue have been included in the revised paper, see **section 2.4** or **Lines 177-199**, as follows:

"Therefore, in this study, the retrieval of the mixing state of BC is based on the assumption that the refractory component in sub micrometre aerosols consists mainly or solely of BC. This assumption might not be true that there are some other important non-volatile aerosol compounds in submicron aerosols, for instance, some extremely low-volatility organic material that does not evaporate even at 300 °C (Cappa & Jimenez, 2010; Häkkinen et al., 2012; Poulain et al.,2014; Zhang et al., 2016; Wang et al., 2017). To investigate the composition of the refractory component and verify whether they consist mainly of BC, we first quantify the mass concentration of these non-volatile material, which was determined from the measured particle number size distributions after heating by assuming particle density of 1.6 g cm$^{-3}$ (Häkkinen et al., 2012; Poulain et al., 2014). Then, we correlate it with the BC mass concentrations measured by AE33, which are shown in Fig. 3. The calculated non-volatile particle mass concentration and the measured BC concentration correlated well, with slope of 1.02. The mass fraction of non-volatile compounds except BC was further evaluated, which accounted for ~1.8 %. Consequently, at 300 °C, contribution of non-volatile OA factors is expected to be quite negligible (< 5 %). This result suggests that BC can explain almost the non-volatile mass fraction in this study. In addition, we compared the mean VSF measured at 200 and 300 °C, results show that the VSF values varied greatly under different heating temperatures especially for large particles (Fig. S2), hence some studies (Xu et al., 2016; Cappa & Jimenez, 2010) obtained considerable non-volatile OA at 250 °C may differ from the fractions at 300 °C."

[Figure]

**Figure R1.** Black carbon (BC) mass concentration measured by AE33 vs. the non-volatile mass concentration estimated from the VTDMA for winter 2019 periods. Estimation of the non-volatile mass concentration was made assuming a density of 1.6 g cm$^{-3}$.

[Figure]

**Figure R2.** The mean volatility shrink factor (VSF) of all measured size particles after heating at 200 (blue line) and 300 ℃ (red line) during the winter and summer periods.

Then, it comes with my second comment regarding the method this manuscript used to category In-BC, Ex-BC and Non-BC particles as given in line 146-149. Even though this method has been used in previous studies as in Wehner et al. (2009) and Cheng et al. (2012), it is actually out of date and not recommended to be employed to classify the mixing state of BC-particles. With VSF > 0.82, ambient particles could be composed of only BC, but also In-BC coating with non-volatile organics. Therefore, using current classification, significant uncertainties may be introduced into the estimation of the number fraction of different BC-containing particles. We now rely on some other techniques, for instance, SP2, VTDMA-SP2, SP-AMS to study the mixing state of BC-containing particles.

**Re:** It is true that currently some advanced techniques such as SP2 and SP-AMS were used widely to investigate the mixing states of BC particles, which is more accurate and intuitive. However, due to the lack of these instruments, and considering various studies have also used the VTDMA to estimate the mixing states of soot particles (Philippin et al., 2004; Rose et al., 2011; Levy et al., 2014; Zhang et al., 2016), so we tried to retrieve the particle mixing state by VTDMA. In previous studies, particles with different volatile fractions (i.e., different VSF values) at 300 °C are often assumed to be soot particles with different mixing states (Cheng et al., 2006; Wehner et al., 2009; Cheng et al., 2012). The further evidence that support this retrieved results in this study is included by comparing the retrieved Ex-BC with that measured by SPAMS data (see the next comment).

Moreover, in line 161-162, you concluded your results were reliable for deriving the mixing state of BC by comparing with SPAMS results. However, Fig.2 in your previous publication (Chen et al., 2020) only compared the total BC-containing particles; separation of Ex-BC particles from In-BC particles was not performed. I did not see any other evidence that could support your conclusion here. If you have, please specify.

**Re:** The reliability of retrieved Ex-BC is also verified by comparison with SPAMS data, see Fig. R3, both the overall temporal variations and proportion for Ex-BC particles measured by SPAMS and calculated from VTDMA using the retrieved method are consistent, confirming our method are reliable for retrieve mixing state of BC in the study periods. We have added this part of evidence to the revised paper, see **Lines 200-208**, as follows:

"To further verify the reliability of the retrieved results, the number fraction of Ex-BC calculated from the VTDMA is compared with the measurements by single particle aerosol mass spectrometer (SPAMS) (Bi et al., 2015), as shown in the Fig. 4. It exhibits that the variations of number fractions of the Ex-BC particles retrieved from VTDMA are well consistent with that measured by SPAMS, confirming that the method is reliable for deriving the mixing state of BC during the campaign in urban Beijing."

[Figure]

**Figure R3.** Time series of number fraction of Ex-BC particles measured by SPAMS (in black) and calculated from VTDMA (in red), the 200 nm particles from VTDMA are chosen.

Large uncertainties using Eq. 1 in your manuscript to calculate completely volatile particles may arise. Transportation efficiency as I understand was determined by volatility measurements of NaCl particles or other non-volatile particles at 300 °C in your study. However, ambient aerosols with different volatility could evaporate differently, resulting in different particle size after heating. Therefore, you have to at least consider the number size distribution of particles after heating into the determination of the transportation efficiency. In other words, number concentration of completely volatile particles should be dependent on the volatility of your ambient particles, thus your Eq.1 should be revised as a function of your VSF or your volatility distribution after heating.

**Re:** The transportation efficiency η was affected by particle diffusional and thermophoretic losses in the sampling lines, and determined at each particle diameter and heating temperature with NaCl particles in laboratory calibrations (Philippin et al., 2004; Cheung et al., 2016), which do not evaporate at the temperature (300 °C) used in our measurements. In this study, we have considered the number size distribution loss of NaCl particles after heating at 300 °C to calculate the η. To describe more detailly, we add an equation to explain the η in line 148, as follows:

$$\eta_{D_p,T} = \frac{N_r(NaCl)}{N_{D_p}(NaCl)} \tag{1}$$

A statement has also been included in the revised version, see **lines 141-143**, as follows:

"In this study, $\eta_{D_p,T}$ at each particle size is determined from the number concentration of sodium chloride (NaCl) particles before and after heating at 300 °C (i.e. $\eta_{D_p,T} = N_r(NaCl)/N_{D_p}(NaCl)$)."

In section 3.3, you studied the effect of the formation and growth of particles on their volatility. However, your discussion and analysis is not thorough. How frequent of NPF events occurred during your studied seasons stated in your manuscript? Did you select the data of the days of NPF events to drive your conclusion, otherwise, how could the

influence of other atmospheric processes be neglected? What are the sources of 40 and 150 nm particles? Your 40 nm particles could be newly formed particles or a mixture of newly formed particles after growth with pre-exsiting particles. Your analysis should guide the reader to a clear process level, otherwise your conclusion will be quite difficult for us to sink into. Moreover, are the plots in Fig. 4a-h the average or median values of the whole campaign or of the NPF days? I think Fig. 4g or Fig. 4h in your manuscript present a certain NPF day, how could you compare general patterns with a case event? How could we reader withdraw general information or conclusions by case studies?

**Re:** ① The NPF events took place frequently during the summer sampling periods, with about 10 NPF events occurred (see Fig. R4). The time series of the aerosol particle number size distribution measured by the SMPS during the campaign have been added to the supplement (Fig. S5).

②The pre-existing 40-nm particles could be much fewer as the number concentration was much lower as compared with the contribution by the growth of nucleated particles (Wang et al., 2018). Therefore, the 40-nm particles during the NPF events can represent the newly formed particles.

③ The plots in Fig. 4a-h are the average values of the whole campaign. After checking the original data, we just have updated the Fig. 4g and 4h, or see Fig. R5.

Some discussions have been included in the revised text of **Lines 281-285** and **Lines 290-295**, as follows:

"Here, the VSF-PDFs of 40- and 150-nm particles are shown. The variation of pre-existing 40-nm particles could be much fewer as the number concentration was much lower as compared with the contribution by the growth of nucleated particles (Wang et al., 2018). Therefore, the 40-nm particles during the NPF events can represent the newly formed particles, and the 150-nm particles represent the pre-existing particles."

"Fig. 8g displays the mean diurnal variations of PNSDs in the summer. During the summer sampling periods, new particle formation (NPF) events took place frequently, with 10 NPF events occurred (Fig. S5). The NPF events almost all started at around 10:00 LT. After the starting of NPF, the volatile mode in VSF-PDF was obviously enhanced, corresponding to significant decreases of the mean VSF values. This suggests that the more volatile chemical components were formed in the nucleation and growth processes."

[Figure]

**Figure R4.** Time series of the aerosol particle number size distribution measured by the SMPS during the **(a)** summer and **(b)** winter periods.

[Figure]

**Figure R5.** Diurnal variation of **(a-b)** mean VSF for all measured dry particle sizes, **(c-f)** mean VSF-PDF for 40- and 150-nm particles, and **(g-h)** mean particle number size distribution during the summer (left) and winter (right) periods. The shade regions in (a-b) denote the standard deviations.

In section 3.6, you characterized coating thickness by DP/DC ratio. However, I did not find any description or definition about Dp and Dc. You obtained a volatility distribution of particles after heating. How did you get Dc from this distribution? The determination of Dc was quite difficult, even with a SP2, which could accurately

measure the mass of BC. Zhang et al. (2016) gave a thorough discussion of the uncertainties associated with determining Dc from SP2 measurements due to the morphology effect. Similar discussion of the uncertainties in determining Dc should be given in the manuscript.

**Re:** Bimodal distribution was observed in VTDMA measurements, representing the size distributions of Ex-BC and In-BC cores respectively. The peaks at initially prescribed size ranges represent Ex-BC particles because they did not undergo a size change after heating. The peaks at smaller size represent the sizes of In-BC cores. An example of the measured size distribution is shown in Fig. R6. Here, $D_p$ refers to the peak value of the $DMA_1$ selected size distribution (particles with diameters of 150 nm are presented as examples), and $D_c$ refers to the peak diameter of residual particles after heating at 300 °C. The heated particles should be more compacted with nearly spherical core, which is expected to be very different from uncoated fresh BC that are with fractural chain morphology. In the revised paper, we have given the definition about $D_p$ and $D_c$ in **Lines 166–168**, as follows:

"Here, particles with diameters of 150 nm are presented as examples (Fig. 4), $D_p$ refers to the peak value of the $DMA_1$ selected size distribution, and $D_c$ refers to the peak diameter of residual particles after heating at 300 ℃."

[Figure]

**Figure R6.** Number size distributions of ambient aerosols (in black), $DMA_1$-selected particles with $D_p$ equal to 150 nm (in blue), residual particles after heating at 300 °C (in red), and the fitting curves.

**Reference**

Birmili, W., Weinhold, K., Nordmann, S., Wiedensohler, A., Spindler, G., Müller, K., Herrmann, H., Gnauk, T., Pitz, M., Cyrys, J., Flentje, H., Nickel, C., Kuhlbusch, T. A. J., and Löschau, G.: Atmospheric aerosol measurements in the German Ultrafifine Aerosol Network (GUAN): Part 1 – soot and particle number size distribution, Gefahrst. Reinh. Luft., 69, 137–145, 2009.

Birmili, W., Heinke, K., Pitz, M., Matschullat, J., Wiedensohler, A., Cyrys, J., Wichmann, H.-E., and Peters, A.: Particle number size distributions in urban air before and after volatilisation, Atmos. Chem. Phys., 10, 4643–4660, doi:10.5194/acp-10-4643-2010, 2010.

Cappa, C. D. and Jimenez, J. L.: Quantitative estimates of the volatility of ambient organic aerosol, Atmos. Chem. Phys., 10, 5409–5424, doi:10.5194/acp-10-5409-2010, 2010. Chen, L., Zhang, F., Yan, P., Wang, X., Sun, L., Li, Y., Zhang, X., Sun, Y., and Li, Z.: The large proportion of black carbon (BC)-containing aerosols in the urban atmosphere, Environmental Pollution, 263, 114507, https://doi.org/10.1016/j.envpol.2020.114507, 2020.

Cheng, Y.F., Eichler, H., Wiedensohler, A., Heintzenberg, J., Zhang, Y.H., Hu, M., Herrmann, H., Zeng, L.M., Liu, S., Gnauk, T., Brüggemann, E., He, L.Y., 2006. Mixing state of elemental carbon and non-light-absorbing aerosol components derived from in situ particle optical properties at Xinken in Pearl River Delta of China. J. Geophys. Res.: Atmosphere 111, D20204.

Cheng, Y.F., Su, H., Rose, D., Gunthe, S.S., Berghof, M., Wehner, B., Achtert, P., Nowak, A., Takegawa, N., Kondo, Y., Shiraiwa, M., Gong, Y.G., Shao, M., Hu, M., Zhu, T., Zhang, Y.H., Carmichael, G.R., Wiedensohler, A., Andreae, M.O., Pöschl, U., 2012. Size-resolved measurement of the mixing state of soot in the megacity Beijing, China: diurnal cycle, aging and parameterization. Atmos. Chem. Phys. 12, 4477e4491.

Cheung, H.H.Y., Tan, H., Xu, H., Li, F., Wu, C., Yu, J.Z., Chan, C.K., 2016. Measurements of non-volatile aerosols with a VTDMA and their correlations with carbonaceous aerosols in Guangzhou, China. Atmos. Chem. Phys. 16, 8431e8446.

Frey, A., Rose, D., Wehner, B., Müller, T., Cheng, Y., Wiedensohler, A., Virkkula, A., 2008. Application of the volatility-TDMA technique to determine the number size distribution and mass concentration of less volatile particles. Aerosol Sci. Technol. 42, 817e828.

Häkkinen, S. A. K., Äijälä, M., Lehtipalo, K., Junninen, H., Back man, J., Virkkula, A., Nieminen, T., Vestenius, M., Hakola, H., Ehn, M., Worsnop, D. R., Kulmala, M., Petäjä, T., and Riipinen, I.: Long-term volatility measurements of submicron atmospheric aerosol in Hyytiälä, Finland, Atmos. Chem. Phys., 12, 10771–10786, doi:10.5194/acp-12-10771-2012, 2012.

Levy, M. E., Zhang, R., Zheng, J., Tan, H., Wang, Y., Molina, L. T., Takahama, S., Russell, L. M., and Li, G.: Measurements of submicron aerosols at the California–Mexico border during the Cal–Mex 2010 field campaign, Atmos. Environ., 88, 308–319, doi:10.1016/j.atmosenv.2013.08.062, 2014.

Philippin, S., Wiedensohler, A., and Stratmann, F.: Measurements of non-volatile fractions of pollution aerosols with an eight-tube volatility tandem differential mobility analyzer (VTDMA-8), Journal of Aerosol Science, 35, 185-203, https://doi.org/10.1016/j.jaerosci.2003.07.004, 2004.

Pöschl, U.: Atmospheric Aerosols: Composition, Transformation, Climate and Health Effects, Angewandte Chemie International Edition, 44, 7520–7540, doi:10.1002/anie.200501122, 2005.

Poulain, L., Birmili, W., Canonaco, F., Crippa, M., Wu, Z. J., Nordmann, S., Spindler, G., Prévôt, A. S. H., Wiedensohler, A., and Herrmann, H.: Chemical mass balance of 300 ◦C non-volatile

particles at thetropospheric research site Melpitz, Germany, Atmos. Chem. Phys., 14, 10145–10162, doi:10.5194/acp-14-10145-2014, 2014.

Rose, D., Gunthe, S. S., Su, H., Garland, R. M., Yang, H., Berghof, M., Cheng, Y. F., Wehner, B., Achtert, P., Nowak, A., Wiedensohler, A., Takegawa, N., Kondo, Y., Hu, M., Zhang, Y., Andreae, M. O., and Pöschl, U.: Cloud condensation nuclei in polluted air and biomass burning smoke near the mega-city Guangzhou, China – Part 2: Size-resolved aerosol chemical composition, diurnal cycles, and externally mixed weakly CCN-active soot particles, Atmos. Chem. Phys., 11, 2817–2836, doi:10.5194/acp-11-2817-2011, 2011.

Wang, X., Shen, X. J., Sun, J. Y., Zhang, X. Y., Wang, Y. Q., Zhang, Y. M., Wang, P., Xia, C., Qi, X. F. and Zhong, J. T.: Size-resolved hygroscopic behavior of atmospheric aerosols during heavy aerosol pollution episodes in Beijing in December 2016, Atmos. Environ., 194, 188–197, doi:10.1016/j.atmosenv.2018.09.041, 2018.

Wang, Z., W. Birmili, A. Hamed, B. Wehner, G. Spindler, X. Pei, Z. Wu, Y. Cheng, H. Su, and A. Wiedensohler: Contributions of volatile and nonvolatile compounds (at 300 °C) to condensational growth of atmospheric nanoparticles: An assessment based on 8.5 years of observations at the Central Europe background site Melpitz, J. Geophys. Res. Atmos., 122, 485–497, doi:10.1002/2016JD025581, 2017.

Wehner, B., Berghof, M., Cheng, Y. F., Achtert, P., Birmili, W., Nowak, A., Wiedensohler, A., Garland, R. M., Pöschl, U., Hu, M., and Zhu, T.: Mixing state of nonvolatile aerosol particle fractions and comparison with light absorption in the polluted Beijing region, 114, 10.1029/2008jd010923, 2009.

Xu, L., Williams, L. R., Young, D. E., Allan, J. D., Coe, H., Massoli, P., Fortner, E., Chhabra, P., Herndon, S., Brooks, W. A., Jayne, J. T., Worsnop, D. R., Aiken, A. C., Liu, S., Gorkowski, K., Dubey, M. K., Fleming, Z. L., Visser, S., Prévôt, A. S. H., and Ng, N. L.: Wintertime aerosol chemical composition, volatility, and spatial variability in the greater London area, Atmos. Chem. Phys., 16, 1139–1160, https://doi.org/10.5194/acp-16-1139-2016, 2016.

Zhang, S. L., Ma, N., Kecorius, S., Wang, P. C., Hu, M., Wang, Z. B., Größ, J., Wu, Z. J., and Wiedensohler, A.: Mixing state of atmospheric particles over the North China Plain, Atmos. Environ., 125, Part A, 152–164, doi:10.1016/j.atmosenv.2015.10.053, 2016.

Zhang, Y., Zhang, Q., Cheng, Y., Su, H., Kecorius, S., Wang, Z., Wu, Z., Hu, M., Zhu, T., Wiedensohler, A., and He, K.: Measuring the morphology and density of internally mixed black carbon with SP2 and VTDMA: new insight into the absorption enhancement of black carbon in the atmosphere, Atmos. Meas. Tech., 9, 1833–1843, https://doi.org/10.5194/amt-9-1833-2016, 2016.

**A point-by-point response to reviewer**

**Reviewer #2**

General Comments:

Chen et al. conducted ambient measurements in Beijing in summer and winter from which they calculated size-dependent volatility shrinkage factors (VSF) and mixing states of urban aerosols and compared the volatility properties in different seasons. The measurements were conducted with the use of a VTDMA of which size-selected aerosols ranging from 40-nm to 300-nm were heated up to 300°C. The non-volatile particles that remained in the particle phase upon heating up to 300°C were assumed to be black carbon (BC) in the analysis. Although volatility analysis of ambient aerosols has been intensively studied in general, this manuscript presents measurement results in different seasons in north China. The dataset presented in the manuscript is overall comprehensive but could be more thorough when interpreting the results.

**Re:** We are grateful to the reviewer for your insightful comments, all of which have been considered carefully during the revision (a point-by-point response to reviewer as follows). The result has been carefully analyzed accordingly to the reviewer's suggestions and other comments have also been addressed.

Specific Comments:

1. I have similar concerns about the major assumption of attributing the non-volatile composition in urban aerosols to be BC in your analysis, as already detailed by another referee. Please consider providing more information, such as chemical composition data, or data from other instruments, if available, to support your assumption, which is critical to your analyses and discussions thereafter.

**Re:** It is possible that there may be some other non-volatile compounds in submicron aerosols will not evaporate even at 300 ℃. To investigate the composition of the refractory component and verify whether they consist mainly of BC, we quantify the mass concentration of these non-volatile material and correlate it with the measured BC mass concentrations by AE33, which are shown in Fig. R1. The calculated non-volatile particle mass concentration and the measured BC concentration correlated well, with slope of 1.02. The mass fraction of non-volatile compounds except BC was further evaluated, which accounted for ~1.8 %. Consequently, at 300 ℃, contribution of non-volatile materials except BC is expected to be quite negligible (< 5 %). This result suggests that BC can explain almost the non-volatile mass fraction in this study. In addition, we compared the mean VSF measured at 200 and 300 ℃, results show that the VSF values varied greatly under different heating temperatures especially for large particles (Fig. R2), hence some studies (Xu et al., 2016; Cappa & Jimenez, 2010) obtained considerable non-volatile OA at 250 ℃ may differ from the fractions at 300 ℃.

Actually, the composition of these non-volatile residuals may vary spatially and temporally (Poulain et al., 2014). For example, black carbon is considered a major non-volatile component in sub-μm PM in many studies (e.g., Pöschl, 2005; Frey et al., 2008; Birmili et al., 2009; Birmili et al., 2010). Frey et al. (2008) have demonstrated a good agreement between the mass concentration of BC and the mass concentration of non-volatile particles upon heating at 300 ℃ in the VTDMA; Birmili et al. (2009) found linear relationships between the BC mass concentrations and the non-volatile volume concentrations for five different atmospheric measurement sites; Birmili et al. (2010) found the non-volatile aerosol material had a clear correlation with BC aerosols in polluted areas. Generally, BC constitutes a major part of the non-volatile mass concentration.

The reliability of retrieved Ex-BC is also verified by comparison with SPAMS data, see Fig. R3, both the overall temporal variations and proportion for Ex-BC particles measured by SPAMS and calculated from VTDMA using the retrieved method are consistent, confirming our method are reliable for retrieving mixing state of BC in the study periods. Some discussions and evidences about this issue have been included in the revised paper, see **section 2.4** or **Lines 172-208**, as follows:

"Therefore, in this study, the retrieval of the mixing state of BC is based on the assumption that the refractory component in sub micrometre aerosols consists mainly or solely of BC. This assumption might not be true that there are some other important non-volatile aerosol compounds in submicron aerosols, for instance, some extremely low-volatility organic material that does not evaporate even at 300 ℃ (Cappa & Jimenez, 2010; Häkkinen et al., 2012; Poulain et al.,2014; Zhang et al., 2016; Wang et al., 2017). To investigate the composition of the refractory component and verify whether they consist mainly of BC, we first quantify the mass concentration of these non-volatile material, which was determined from the measured particle number size distributions after heating by assuming particle density of 1.6 g cm$^{-3}$ (Häkkinen et al., 2012; Poulain et al., 2014). Then, we correlate it with the BC mass concentrations measured by AE33, which are shown in Fig. 3. The calculated non-volatile particle mass concentration and the measured BC concentration correlated well, with slope of 1.02. The mass fraction of non-volatile compounds except BC was further evaluated, which accounted for ~1.8 %. Consequently, at 300 ℃, contribution of non-volatile OA factors is expected to be quite negligible (< 5 %). This result suggests that BC can explain almost the non-volatile mass fraction in this study. In addition, we compared the mean VSF measured at 200 and 300 ℃, results show that the VSF values varied greatly under different heating temperatures especially for large particles (Fig. S2), hence some studies (Xu et al., 2016; Cappa & Jimenez, 2010) obtained considerable non-volatile OA at 250 ℃ may differ from the fractions at 300 ℃.

To further verify the reliability of the retrieved results, the number fraction of Ex-BC calculated from the VTDMA is compared with the measurements by single particle aerosol mass spectrometer (SPAMS) (Bi et al., 2015), as shown in the Fig. 4. It exhibits that the variations of number fractions of the Ex-BC particles retrieved from VTDMA are well consistent with that measured by SPAMS, confirming that the method is

reliable for deriving the mixing state of BC during the campaign in urban Beijing."

[Figure]

**Figure R1.** Black carbon (BC) mass concentration measured by AE33 vs. the non-volatile mass concentration estimated from the VTDMA for winter 2019 periods. Estimation of the non-volatile mass concentration was made assuming a density of 1.6 g cm⁻³.

[Figure]

**Figure R2.** The mean volatility shrink factor (VSF) of all measured size particles after heating at 200 (blue line) and 300 ℃ (red line) during the winter and summer periods.

[Figure]

**Figure R3.** Time series of number fraction of Ex-BC particles measured by SPAMS (in black) and calculated from VTDMA (in red), the 200 nm particles from VTDMA are chosen.

2. Since the manuscript aims to characterize the volatility properties of urban aerosols and tries to link the properties to the source, formation and growth, the authors may consider adding more materials to enrich the discussion, such as air masses origins information and their effect on the aerosols' volatility and mixing states.

**Re:** According to the reviewer's suggestion, we have added the air masses origins information during the two sampling periods (Fig. R4), and some discussions about their effect on the aerosols' volatility and mixing state have been included in the revised paper, see **Lines 257-277**, as follows:

"Fig. 7 presents the 72-h back trajectories arriving at the sampling site during the two periods from 00:00 to 23:00 LT calculated applying the TrajStat software (Wang et al., 2009) (Fig. 7a), and the size-resolved mean VSF (VSF$_{mean}$) of the corresponding cluster during the winter and summer periods (Fig. 7b). In the winter, the air masses were categorized into five clusters and with prevailing northerly wind. The northwest clusters (C2 and C3) were predominant, which associated with the high PM$_{2.5}$ concentrations (Wang et al., 2015). The VSF$_{mean}$ difference of small particle size among different clusters is greater than that of large size, implying the more diverse sources of small particles. In the summer, the air masses were classified into six clusters, with prevailing southerly wind (C1 and C3). It shows that the VSF$_{mean}$ values in both the winter and summer are not closely associated with the variations of trajectories, or the impacts of regional transportation on volatility of the fine aerosol particles are complex. Obviously, the seasonal differences in VSF$_{mean}$ are more significant than that among different clusters especially for larger size particles, e.g. with lower VSF$_{mean}$ value of ~0.47 in the summer than that in the winter (~0.55)."

[Figure]

**Figure R4. (a)** The 72-h back trajectories arriving at Nanjiao site during the winter (left) and summer (right) periods. C1-C6 represent Cluster 1-Cluster 6 respectively. The percentages present the relative occurrences. **(b)** Box diagram for the mean volatility shrink factor (VSF$_{mean}$) of all selected diameter particles (40-300 nm) from different clusters during the winter (blue) and summer (red) periods. The horizontal line in the block diagram represents the median, the diamond represents the mean, the upper and lower borders represent the 25th and 75th percentiles, and the upper and lower borders of the dotted vertical line represent the 10th and 90th percentiles.

3. It is frequently mentioned throughout Section 3 about the impact of new particle formation (NPF) and the growth on the volatility properties of aerosols. Please consider providing more details, such as the number of NPF events in summer and winter, respectively, to give a clearer picture and support to your analysis. Furthermore, although NPF events occurred less frequently in winter, did it have similar impact on aerosols' volatility as that in summer?

**Re:** The NPF events took place frequently during the sampling periods of summer, with about 10 NPF events occurred (see Fig. R5). The time series of the aerosol particle number size distribution measured by the SMPS during the campaign have been added to the supplement (Fig. S5). Fig. R5 shows that almost no NPF events occurred during the campaign. Some statements have been included in the revised text of **Lines 290-295**, as follows:

"Fig. 8g displays the mean diurnal variations of PNSDs in the summer. During the summer sampling periods, new particle formation (NPF) events took place frequently, with 10 NPF events occurred (Fig. S5). The NPF events almost all started at around 10:00 LT. After the starting of NPF, the volatile mode in VSF-PDF was obviously enhanced, corresponding to significant decreases of the mean VSF values. This suggests that the more volatile chemical components were formed in the nucleation and growth processes."

[Figure]

**Figure R5.** Time series of the aerosol particle number size distribution measured by the SMPS during the **(a)** summer and **(b)** winter periods.

4. In line 178 – 181, the authors mention that the distributions of VSF for 150-nm particles were generally unimodal in both summer and winter. However, from Fig. 2(e), it seems that 150-nm particles were generally bi-modal with a non-volatile mode and a high-volatile mode in winter.

**Re:** Thanks for the comments. The description has been revised in **Lines 223-227**, as follows:

"For the 150-nm particles, the distributions were generally unimodal, with VSF of about 0.3-0.6 in the winter, but were almost bimodal with a non-volatile mode and a high-volatile mode in the summer, indicating the mixing and aging of the primary particles during growth to larger sizes during the winter sampling periods."

5. Section 3.3 compares the diurnal variation of particles volatility between summer and winter based on the mean VSF and VSF probability distribution function (VSF-PDF) as illustrated on Fig. 4, yet I was lost from line 229 to 236 when the number fraction of the low-volatile mode is discussed. Is the discussion still based on Fig. 4 or other figures in the manuscript?

**Re:** Yes, the discussion is still based on Fig. 4c-f. Actually, the less-volatile mode for both 40- and 150-nm particles was more significant during the summer. A statement has been included in the revised text, see **Lines 313,** as follows:

"In addition, the number fraction of the LV mode for both 40- and 150-nm particles is much lower during the winter (Figs. 8c-f)."

6. Fig. 5(a) presents the time series of the number concentrations of Non-BC, In-BC and Ex-BC 150-nm particles in summer and winter. While this work shall be the same as that presented in their previous publication (Chen et al., 2020), the number concentration of 150-nm particles in this manuscript seems to be different from that on Fig. 5(a) in Chen et al. (2020). The scale also looks different from that of other sizes as shown in Fig. S4 in the Supplement.

**Re:** Thank you for your careful check. Per your comments, we checked the data and found some are incorrect that used in Fig. 5(a) in this manuscript. We have updated the figure, as follows (see Figs. R6 and R7):

[Figure]

**Figure R6.** Summer (top) and winter (bottom) time series of **(a)** number concentrations and **(b)** number fractions of Non-BC (in green), In-BC (in blue), and Ex-BC (in red) 150-nm particles.

[Figure]

**Figure R7.** Temporal variation of number concentrations of Non-BC (in green), In-BC (in blue), and Ex-BC (in red) in the range of 40-300 nm particles during the winter periods.

7. In line 285 – 288, the authors state: "non-BC concentration / fraction in winter exhibits a daily minimum and nightly maximum". It is not clear to me whether this observation is supported by Fig. 7. For example, for 40-nm particles, there seems to be a morning peak at 08:00 LT for non-BC concentration in winter. For 150-nm particles, I am not sure whether there was any significant diurnal cycle for non-BC concentration / fraction. Please further elaborate to support your analysis.

**Re:** Yes, the diurnal cycle of Non-BC concentration for 150-nm particles is less obvious than that for 40-nm particles. A slight increase for both the number concentration and fraction can be found for 150-nm particles during nighttime. The average Non-BC number concentration was ~1470 cm$^{-3}$ during nighttime (21:00-05:00 LT), and the number fraction was ~40 %, while the average Non-BC concentration during daytime (06:00-20:00 LT) was ~942 cm$^{-3}$, and the corresponding fraction was ~30 %, showing Non-BC concentration/fraction was lower during daytime and higher during nighttime. According to the reviewer's suggestion, we have added more detailed descriptions in the revised version, see **Lines 366-369**, as follows:

"Compared to the summer, the Non-BC concentration/fraction in the winter was lower during daytime and higher during nighttime, with the average number concentration of ~1470 cm$^{-3}$ during nighttime (21:00-05:00 LT), corresponding to number fraction of ~40 %, but only ~942 cm$^{-3}$ (~30%) during daytime (06:00-20:00 LT)."

8. The analysis method for the ratio of BC diameter discussed in Section 3.6 should be added in Section 2.

**Re:** Revised.

**Reference**

Birmili, W., Weinhold, K., Nordmann, S., Wiedensohler, A., Spindler, G., Müller, K., Herrmann, H., Gnauk, T., Pitz, M., Cyrys, J., Flentje, H., Nickel, C., Kuhlbusch, T. A. J., and Löschau, G.: Atmospheric aerosol measurements in the German Ultrafifine Aerosol Network (GUAN): Part 1 – soot and particle number size distribution, Gefahrst. Reinh. Luft., 69, 137–145, 2009.

Birmili, W., Heinke, K., Pitz, M., Matschullat, J., Wiedensohler, A., Cyrys, J., Wichmann, H.-E., and Peters, A.: Particle number size distributions in urban air before and after volatilisation, Atmos. Chem. Phys., 10, 4643–4660, doi:10.5194/acp-10-4643-2010, 2010.

Cappa, C. D. and Jimenez, J. L.: Quantitative estimates of the volatility of ambient organic aerosol, Atmos. Chem. Phys., 10, 5409–5424, doi:10.5194/acp-10-5409-2010, 2010. Chen, L., Zhang, F., Yan, P., Wang, X., Sun, L., Li, Y., Zhang, X., Sun, Y., and Li, Z.: The large proportion of black carbon (BC)-containing aerosols in the urban atmosphere, Environmental Pollution, 263, 114507, https://doi.org/10.1016/j.envpol.2020.114507, 2020.

Frey, A., Rose, D., Wehner, B., Müller, T., Cheng, Y., Wiedensohler, A., Virkkula, A., 2008. Application of the volatility-TDMA technique to determine the number size distribution and mass concentration of less volatile particles. Aerosol Sci. Technol. 42, 817e828.

Häkkinen, S. A. K., Äijälä, M., Lehtipalo, K., Junninen, H., Back man, J., Virkkula, A., Nieminen, T., Vestenius, M., Hakola, H., Ehn, M., Worsnop, D. R., Kulmala, M., Petäjä, T., and Riipinen, I.: Long-term volatility measurements of submicron atmospheric aerosol in Hyytiälä, Finland, Atmos. Chem. Phys., 12, 10771–10786, doi:10.5194/acp-12-10771-2012, 2012.

Pöschl, U.: Atmospheric Aerosols: Composition, Transformation, Climate and Health Effects, Angewandte Chemie International Edition, 44, 7520–7540, doi:10.1002/anie.200501122, 2005.

Poulain, L., Birmili, W., Canonaco, F., Crippa, M., Wu, Z. J., Nordmann, S., Spindler, G., Prévôt, A. S. H., Wiedensohler, A., and Herrmann, H.: Chemical mass balance of 300 ∘C non-volatile particles at thetropospheric research site Melpitz, Germany, Atmos. Chem. Phys., 14, 10145–10162, doi:10.5194/acp-14-10145-2014, 2014.

Wang, L., Liu, Z., Sun, Y., Ji, D., and Wang, Y.: Long-range transport and regional sources of PM2.5 in Beijing based on long-term observations from 2005 to 2010, Atmospheric Research, 157, 37-48, https://doi.org/10.1016/j.atmosres.2014.12.003, 2015.

Wang, Y., Zhang, X., Draxler, R.R., 2009. TrajStat: GIS-based software that uses various trajectory statistical analysis methods to identify potential sources from long-term air pollution measurement data. Environ. Model. Software 24, 938–939.

Xu, L., Williams, L. R., Young, D. E., Allan, J. D., Coe, H., Massoli, P., Fortner, E., Chhabra, P., Herndon, S., Brooks, W. A., Jayne, J. T., Worsnop, D. R., Aiken, A. C., Liu, S., Gorkowski, K., Dubey, M. K., Fleming, Z. L., Visser, S., Prévôt, A. S. H., and Ng, N. L.: Wintertime aerosol chemical composition, volatility, and spatial variability in the greater London area, Atmos. Chem. Phys., 16, 1139–1160, https://doi.org/10.5194/acp-16-1139-2016, 2016.

---

## Author Response (AR2)

**A point-by-point response to reviewer**

Dear Editor,

   We are very pleased to submit a revised manuscript entitled with "Characterizing the volatility and mixing state of ambient fine particles in summer and winter of urban Beijing" for possible publication in journal of Atmospheric Chemistry and Physics.

   We'd like to thank you for your efforts and time on handling the paper. We also thank the reviewer 1 for the further comments, which we have addressed in the revision (a point-by-point response to the reviewer as follows).

Yours sincerely,

Fang Zhang

On behalf of all authors

Comments from the reviewer 1:

Even though the author made some revisions to their manuscript and answered some of my comments, I still have some doubts with respect to their answers and hope the authors could clarify:

1. Fig 3 in your revised manuscript. How could you compare the bulk BC mass concentration with size-resolved one, as VTDMA measured the size-resolved VSF. And you obtained a slope of 1.02, which makes you results even suspicious. The bulk one should be larger than the size-resolved one, as the mode size of BC-containing particles should be larger than 300 nm.

**Re:** Thanks for the comments. We are afraid that the method for calculating the bulk mass concentration of the non-volatile material in this study has not been addressed clearly. Therefore, in the revision, we have included more details to clarify this, see **Lines 173-181**, or as follows,

   "…To investigate the composition of the refractory component and verify whether they consist mainly of BC, we first quantify the bulk mass concentration of these non-volatile material. For the calculation, the number concentrations of the residual non-volatile particles at each size (40, 80, 110, 150, 200 and 300 nm) is calculated by integrating the residual PNSD of each selected particle size that directly measured by VTDMA at the temperature of 300 °C. Then, the size resolved mass concentration of

the residual non-volatile particles was calculated by assuming the particles are spherical and with a density of 1.6 g cm$^{-3}$ (Häkkinen et al., 2012; Poulain et al., 2014). Finally, by fitting the size-resolved mass concentration and integrating the fitted curves, the bulk mass concentration of non-volatile particles was retrieved…"

2. Similar as figure 4. Did you use size-resolved information from SPAMS, otherwise how did you compare this two information? There should be some difference, otherwise it is erroneous. Please clarify.

**Re:** According to the reviewer's comments, we have updated and compared the results at 200 nm from both the VTDMA and SPAMS instruments (Fig. R1) in the revised text, see **Lines 194-205**, or as follows,

"…To further verify the reliability of the retrieved results, the number fraction of Ex-BC and In-BC for 200 nm particles calculated from the VTDMA is compared with the measurements by single particle aerosol mass spectrometer (SPAMS), as shown in the Fig. 4. The comparison can only be confined to the size of 200 nm because which is the lower limit of the measured size for SPAMS (Bi et al., 2015). It exhibits that the variations of number fractions of both the Ex-BC and In-BC particles retrieved from VTDMA are well consistent with that measured by SPAMS, confirming that the method is reliable for deriving the mixing state of BC during the campaign in urban Beijing…"

[Figure]

**Figure R1.** Time series of number fraction of **(a)** Ex-BC particles and **(b)** In-BC particles measured by SPAMS (in black) and calculated from VTDMA (in red), the 200 nm particles from VTDMA and SPAMS are chosen for comparison.

3. Since SPAMS could measure the mixing state of BC-containing particles, which is

probably more accurate than VTDMA, what is the reason using VTDMA instead of SPAMS in current study? Your study is neither introducing a new method (actually out of date in my opinion), nor presenting a new result (actually I see you have published half of them in Chen et al. (2020)). What is the reason to use these data again? Then you should emphasis the contribution or objectives for current dataset, for instance, modifying your intro part as well as the result structure.

**Re:** Thanks a lot for the comments.

We agree the reviewer that the SPAMS could measure the mixing state of BC-containing particles, which is probably more accurate than VTDMA. But, the dataset of VTDMA can be used to investigate the aerosol volatility and link the volatility to the particle's formation and growth during cold and warm seasons. In addition, because only a short time period of synchronous data from SPAMS and VTDMA was obtained during the winter campaign, we just used the SPAMS data to verify the inverted mixing state of BC particles. Also, because of restrictions on the data sharing terms, we cannot obtain the right to use and analyze these data from SPAMS more deeply. As the reviewer stated, SPAMS is probably more accurate than VTDMA. In the future, if more dataset from SPAMS is available, further study and analysis can be done.

In this study, we mainly present the size-dependent VSF to characterize and contrast the volatility behavior of fine particles between cold and warm seasons of urban Beijing. The mixing state of ambient fine particles, which is retrieved from the size-resolved VSF, is further analyzed and compared between the two seasons with an aim of understanding of the effects of particle growth on volatility and mixing state in different atmospheric conditions. While Chen et al (2020) only focused on retrieving and quantifying the mass and number concentrations of BC particles with different mixing states. In addition, the data used in this study includes measurements from both summer and winter, while in Chen et al (2020), only winter data were analyzed. More importantly, this study is with new findings that the non-BC particles that primarily from nucleation processes accounted for 52–69 % of the total number concentration in the summer. This thus leads to more volatile particles in the summer than in the winter. This is further evidenced from the diurnal cycles of the retrieved aerosol mixing state and an outstanding high-volatile mode around noontime on new particle formation (NPF) days in summer. While Chen et al (2020) found that large proportions of aerosols are BC-containing particles in the winter of urban Beijing, implying the dominant role of BC in increasing aerosol loadings due to their rapid aging in polluted urban area.

According to the reviewer's suggestion, we just have included some more statements to clarify and emphasis the contribution or objectives for current dataset in the main text and conclusion results (**Lines 77-85, 412-430** or as follows).

"…Here, we used the size-dependent VSF as a parameter to characterize the volatility behavior of fine particles in cold and warm seasons of urban Beijing; In addition, the mixing state of ambient fine particles in the summer, which is retrieved

from the size-resolved VSF, is also compared to that in the winter. By contrasting the volatility and mixing state in the two seasons, this study is with aim of linking the aerosol particles volatile properties and mixing state to its atmospheric chemical and physical processes under different ambient conditions in polluted urban areas…"

"…In this study, the volatility of the fine particles is characterized as VSF and the results from wintertime and summertime are shown and compared. Results show that the measured VSF-PDF is almost always bimodal, with one high-volatile and one less- or non-volatile mode, both in the summer and winter. The mean VSF-PDF has a pronounced HV mode in the summer, generally with a VSF minimum at ~0.2, while in the winter a broad MV mode spans much of the measured VSF range (with VSF minimum at 0.45-0.65), reflecting lower average volatility in the winter. Diurnal variations in VSF of 40-nm particles are evident only in the summer, with a prominent HV mode resulting from new particle formation and growth present around noontime and early afternoon and a dominant LV mode during nighttime. No such feature is evident in the winter measurements. The mixing state of ambient fine particles was calculated from the size-resolved VSF and the results for summer and winter compared and contrasted. On average, nucleation and particle growth results in a dominant population of volatile, Non-BC, particles, which account for 52–69 % of the total concentration in the measured size range in the summer. However, black carbon (BC)-containing particles contributed 67–77 % toward the total number concentration in the winter, indicating that BC particles are a dominant component of the wintertime Beijing. The diurnal cycles of the retrieved particle mixing state in the summer further show that rapid photochemical processing and nucleation are the main contributors to Non-BC particles. By analyzing the ratio of the BC particle diameter before and after heating at 300 ºC, we show that the BC particles are more thickly coated in wintertime than in summertime. The observed results on aerosols volatility and mixing state in this study can help understanding the formation and growth of fine particles and determining their effect on environment and climate…"

4. The discussion part for Sect 3.3 is too weak without sufficient comparison with other studies that deal with particle nucleation and growth. Thus only quite limited information could be extracted for current study.

**Re:** More discussions have been included in the revised paper, see **Lines 283-293**, as follows:

"…After the starting of NPF, the volatile mode in VSF-PDF was obviously enhanced, corresponding to significant decreases of the mean VSF values. This suggests that the more volatile chemical components were formed in the nucleation and growth processes during the campaign. It has been shown that ~97 % of newly formed particles are volatile because they are dominated by non-refractory sulfate and organics in Beijing (Wehner et al., 2009). Wu et al (2017) also observed that a clear decrease in VSFs for 30- and 50-nm particles in rural area of north China during the NPF events,

indicating that more volatile compounds could be produced during the growth process of newly formed particles. However, some earlier campaign measurements that were conducted in various atmospheric environments, such as urban (Sakurai et al., 2005), and forest (Ehn et al., 2007), showed that the volatility of newly formed particles varied with the atmospheric environments, indicating distinct particle growth mechanisms...."

**Reference**

Bi, X., Dai, S., Zhang, G., Qiu, N., Li, M., Wang, X., Chen, D., Peng, P.a., Sheng, G., Fu, J., Zhou, Z., 2015. Real-time and single-particle volatility of elemental carbon-containing particles in the urban area of Pearl River Delta region, China. Atmos. Environ. 118, 194-202.

Ehn, M., Petäjä, T., Birmili, W., Junninen, H., Aalto, P., and Kulmala, M.: Non-volatile residuals of newly formed atmospheric particles in the boreal forest, Atmos. Chem. Phys., 7, 677-684, 10.5194/acp-7-677-2007, 2007.

Häkkinen, S. A. K., Äijälä, M., Lehtipalo, K., Junninen, H., Back man, J., Virkkula, A., Nieminen, T., Vestenius, M., Hakola, H., Ehn, M., Worsnop, D. R., Kulmala, M., Petäjä, T., and Riipinen, I.: Long-term volatility measurements of submicron atmospheric aerosol in Hyytiälä, Finland, Atmos. Chem. Phys., 12, 10771–10786, doi:10.5194/acp-12-10771-2012, 2012.

Poulain, L., Birmili, W., Canonaco, F., Crippa, M., Wu, Z. J., Nordmann, S., Spindler, G., Prévôt, A. S. H., Wiedensohler, A., and Herrmann, H.: Chemical mass balance of 300 ◦C non-volatile particles at thetropospheric research site Melpitz, Germany, Atmos. Chem. Phys., 14, 10145–10162, doi:10.5194/acp-14-10145-2014, 2014.

Sakurai, H., Fink, M. A., McMurry, P. H., Mauldin, L., Moore, K. F., Smith, J. N., and Eisele, F. L.: Hygroscopicity and volatility of 4–10 nm particles during summertime atmospheric nucleation events in urban Atlanta, Journal of Geophysical Research: Atmospheres, 110, https://doi.org/10.1029/2005JD005918, 2005.

Wu, Z. J., Ma, N., Größ, J., Kecorius, S., Lu, K. D., Shang, D. J., Wang, Y., Wu, Y. S., Zeng, L. M., Hu, M., Wiedensohler, A., and Zhang, Y. H.: Thermodynamic properties of nanoparticles during new particle formation events in the atmosphere of North China Plain, Atmospheric Research, 188, 55-63, https://doi.org/10.1016/j.atmosres.2017.01.007, 2017.

---

## Author Response (AR3)

**A point-by-point response to editor**

Dear Editor,

We are very pleased to submit a revised manuscript entitled with "Characterizing the volatility and mixing state of ambient fine particles in summer and winter of urban Beijing" for possible publication in journal of Atmospheric Chemistry and Physics.

We greatly appreciate your effort and time on handling this paper. We also thank you for the further comments, which help us greatly improve this paper.

Yours sincerely,

Fang Zhang

On behalf of all authors

Comments from the editor:

Thank you for the next version of the manuscript! Based on the referee reports, Referee #2 was satisfied with your answers and edits already earlier. After assessing your latest version, I think you have addressed also the concerns of the Referee #1. I have marked some of my own comments in the attached pdf. Some of my suggestions are editorial to improve clarity, some related to terminology and methods while many are related to the role of oxidized organics and their role in aerosol formation, growth and their influences on aerosol volatility.

Line 15: insert the studied sizes here.
**Re:** Revised.

Line 15: that there are two persistent aerosol volatility modes (high-volatile and one less or non-volatile mode) present both in the summer and winter.
**Re:** Revised.

Line 17: This and the next sentence seem to be in contradiction. Figure 6 shows that the summer aerosol in all sizes are more volatile with a lower VSF at smaller sizes. However, the same figure also shows that there is a clear non-volatile mode at 40 nm during the summer. This has actually even higher contribution in the summer than in the winter in the size range pertinent to NPF. The overall process of NPF requires low-volatile vapors to form, cluster and nucleate. It is not feasible that they would be easily evaporated from the particle phase. This is in line with your next sentence.

**Re:** This sentence has been revised, as follows:

"Although the new particle formation (NPF) process requires low-volatile vapors to form molecular clusters and nuclei, the significant high-volatile mode around noontime on NPF days indicates volatile substances could be produced during the growth stage of newly formed particles in the summer."

Line 21: On the other hand.
**Re:** Revised.

Line 70: sources and chemical composition of aerosol particles.
**Re:** Revised.

Line 75: There are also other non-volatile components in addition to BC.
**Re:** This sentence has been revised to "…leaving refractory materials such as BC and other non-volatile components…"

Line 89: there.
**Re:** Revised.

Line 113: was.
**Re:** Revised.

Line 117: How? Please explain. Assuming no evaporation at low temperature and dissociation at a high enough temperature?
**Re:** During the measurement, ambient aerosols were first sampled by a $PM_{2.5}$ inlet and subsequently passed through a Nafion dryer that reduced the sample flow RH to below 30%. "The RH was calibrated periodically with ammonium sulfate during the measurement period" means the accuracy of RH was verified by comparing the delirium point of ammonium sulfate measured by HTDMA with the theoretical delirium point. An explanation has been included in lines 122-124, as follows:

"The accuracy of humidity can be verified by comparing the deliquescence point of ammonium sulfate measured by HTDMA with the theoretical deliquescence point."

Line 123: how?
**Re:** The calibration of VTDMA can be found in lines 116-118: "The VSF-PDF was retrieved based on the $TDMA_{inv}$ algorithm developed by Gysel et al. (2009). The scans in which the temperature between the two DMAs was not increased were used to define the width of the transfer function". The calibration of other instruments (AE-33 and SMPS) can be found Wu et al (2020). A supplementary explanation has been included in line 129: "Detailed calibration process of these auxiliary instruments can be found in Wu et al (2020)".

Line 136: Did you account for the cubic shape factor of NaCl?
**Re:** $\eta_{D_p,T}$ is the transportation efficiency of the sampled particles, which represents

particle losses between DMA₁ and DMA₂ due to diffusion and thermophoretic forces (Philippin et al., 2004), and always determined at each particle diameter and heating temperature with NaCl particles in laboratory calibrations for that they do not evaporate even at high temperatures (Philippin et al., 2004; Cheung et al., 2016). Thus in this study, $\eta_{D_p,T}$ at each particle size is determined from the number concentration of NaCl particles before and after heating at 300 °C (i.e. $\eta_{D_p,T} = N_r(NaCl)/N_{D_p}(NaCl)$). An explanation has been included in lines 140-142, as follows:

"…and always determined at each particle diameter and heating temperature with sodium chloride (NaCl) particles in laboratory calibrations for that they do not evaporate even at high temperatures (Philippin et al., 2004; Cheung et al., 2016)…"

Line 163: split this section into 2-3 paragraphs for clarity and to improve readability.
**Re:** Revised.

Line 202: add the year. is the date MM/DD or DD/MM?
**Re:** Revised.

Line 226: Please make the figure larger to reduce overlap between the labels and to improve clarity.
**Re:** Revised.

Line 245: See my comment in the abstract. Furthermore, organic condensation is highly size dependent. See e.g. Riipinen et al. (2012). For different sized aerosol particles, their growth is governed by organics of different volatility. For example, the semi-volatile organics only condense onto the largest particle sizes whereas the extremely low volatile organics can condense even below 10 nm in size. The prominent HV mode in summer could be due to condensation of semi-volatile material, but this does not have anything to do with new particle formation.
**Re:** This sentence has been revised to "…while the prominent HV mode in the summer especially for large particle size could be due to condensation of semi-volatile materials on nucleated particles (Riipinen et al., 2012)."

Line 266: see my earlier comment regarding organic condensation.
**Re:** This sentence has been revised to "…Obviously, the seasonal differences in VSF_mean are more significant than that of among different clusters especially for larger size particles."

Line 272: There is a considerable tail of BC from traffic that can contribute to the non-volatile fraction in urban environments also at 40 nm size range. Therefore the comparison between 40 nm and 150 nm sizes is far from trivial.
**Re:** Thanks for this comment. It's true that part of the 40 nm particles could be pre-existing particles. The statements have been revised, as follows:

"…Although the local primary emission sources (e.g. vehicles, cooking) can contribute to the pre-existing background small particles in urban environments, the fraction was probably lower during the new particle formation (NPF) events. It was

reported that the ultrafine particles during clear days (e.g. NPF days) in urban Beijing were primarily from secondary formation (Wang et al., 2018), and the pre-existing particles are predominant in accumulated mode."

Line 275: for which size?
**Re:** The size information has been included in this sentence, see line 287, or as follows:
"During the summer, low VSF during the daytime (08:00–18:00 LT) and high VSF during the nighttime were observed for 40 nm and 150 nm particles (Fig. 8a)."

Lines 283-288: see the previous comment.
**Re:** The editor is right about the explanations of decreased VSFs during the NPF events. The discussions have been revised, as follows:
"…Although the NPF process requires low-volatile vapors to form molecular clusters and nuclei (Ehn et al., 2007), the significant high-volatile mode around noontime on NPF days indicates volatile substances could be formed during the growth stage of newly formed particles in the summer. Previous study also showed that the growth process of the nucleated particles primarily formed non-refractory sulfate and organics in Beijing (Wehner et al., 2009)..."

Line 308: Please clarify. Biogenic precursor concentrations are expected to be higher, anthropogenic precursors could be lower. The boundary layer and mixing / dilution can be different.
**Re:** Revised. Here we meant that the anthropogenic precursors could be lower.

Line 339: Pyrolysis can transform the organic aerosol into non-volatile, but this is only after treatment in 300 degC. The "charred" organics probably do not contribute to the growth in ambient conditions.
**Re:** Revised.

Line 354: Please split this section into 2-2 paragraphs to improve readability and clarity.
**Re:** Revised.

Line 412: Please check the conclusions after considering my comments in the earlier sections.
**Re:** Thanks for the editor's constructive comments and suggestions. A revised conclusion has been included in the new version.

---

## Author Response (AR4)

**A point-by-point response to editor**

Dear Editor,

We greatly appreciate your effort and time on handling this paper. The further comments have been considered carefully during the revision (a point-by-point response to editor as follows).

Yours sincerely,

Fang Zhang

On behalf of all authors

Comments from the editor:

Abstract: Although the new particle formation (NPF) process requires low-volatile vapors to form molecular clusters and nuclei, there was a significant high-volatile mode around noontime on NPF days. This indicates partitioning of volatile substances into the growing particles during summer.

**Re:** This statement in the abstract and Section 3.3 has been revised, as follows:

"…the significant high-volatile mode around noontime on NPF days indicates partitioning of volatile substances into the growing particles during summer."

line 125: regarding RH verification with deliquescence: I suggest to use the word "verified" instead of "calibrated" here. Furthermore, the next sentence should state that you verified the humidity conditions, not that it can be done. What was the result of the verification? How much RH varied?

**Re:** Thanks for the editor's suggestion, which has been revised in lines 124-127, or as follows:

"…The result of the verification is shown in Fig. S2. It shows that the deliquescence RH of ammonium sulfate measured by H/V-TDMA is ~78 %, which is consistent with the results reported by Badger et al (2006) and Tan et al (2013), indicating the RH measurement of this system is accurate."

[Figure]

**Figure R1.** The hygroscopic growth factor (GF) of 150 nm $(NH_4)_2SO_4$ particles in different relative humidities (RH). The sudden jump in the GFs near RH 80% denotes the deliquescence RH.

line 145. You did not address my question regarding the shape factor of NaCl in your response. It seems to me that the cubic shape of sodium chloride was not taken into account. Laboratory generated NaCl particles have a cubic shape with a shape factor of 1.08 while the typical assumption is that the non-volatile residual particles are spherical.

**Re:** Thanks for the editor's reminding. After checking the calibration process, the cubic shape factor 1.08 of NaCl particles has been considered in the Igor software. A statement has been included in lines 147-148, or as follows:

"…Due to the cubic shape of NaCl particles, a shape factor of 1.08 was used in the calibration process (Park et al., 2009; Hakala et al., 2017)."

Badger, C. L., George, I., Griffiths, P. T., Braban, C. F., Cox, R. A., and Abbatt, J. P. D.: Phase transitions and hygroscopic growth of aerosol particles containing humic acid and mixtures of humic acid and ammonium sulphate, Atmos. Chem. Phys., 6, 755-768, 10.5194/acp-6-755-2006, 2006.

Hakala, J., Mikkilä, J., Hong, J., Ehn, M., and Petäjä, T.: VH-TDMA: A description and verification of an instrument to measure aerosol particle hygroscopicity and volatility, Aerosol Science and Technology, 51, 97-107, 10.1080/02786826.2016.1255712, 2017.

Park, K., Kim, J.-S., and Miller, A. L.: A study on effects of size and structure on hygroscopicity of nanoparticles using a tandem differential mobility analyzer and TEM, Journal of Nanoparticle Research, 11, 175-183, 10.1007/s11051-008-9462-4, 2009.

Tan, H., Xu, H., Wan, Q., Li, F., Deng, X., Chan, P. W., Xia, D., and Yin, Y.: Design and Application of an Unattended Multifunctional H-TDMA System, Journal of Atmospheric and Oceanic Technology, 30, 1136-1148, 10.1175/jtech-d-12-00129.1, 2013.